# Common dietary emulsifiers promote metabolic disorders and intestinal microbiota dysbiosis in mice
Suraphan Panyod [1,2,3,14], Wei-Kai Wu [3,4,5,6,14], Chih-Ting Chang[1], Naohisa Wada [7], Han-Chen Ho [8], Yi-Ling Lo[3], Sing-Ping Tsai[4], Rou-An Chen[1], Huai-Syuan Huang[1], Po-Yu Liu [3,9], Yi-Hsun Chen[3], Hsiao-Li Chuang[10], Ting-Chin David Shen[11], Sen-Lin Tang [7], Chi-Tang Ho[12], Ming-Shiang Wu [3,5] ✉ & Lee-Yan Sheen [1,2,13] ✉

Dietary emulsifiers are linked to various diseases. The recent discovery of the role of gut microbiota–host interactions on health and disease warrants the safety reassessment of dietary emulsifiers through the lens of gut microbiota. Lecithin, sucrose fatty acid esters, carboxymethylcellulose (CMC), and mono- and diglycerides (MDG) emulsifiers are common dietary emulsifiers with high exposure levels in the population. This study demonstrates that sucrose fatty acid esters and carboxymethylcellulose induce hyperglycemia and hyperinsulinemia in a mouse model. Lecithin, sucrose fatty acid esters, and CMC disrupt glucose homeostasis in the in vitro insulin-resistance model. MDG impairs circulating lipid and glucose metabolism. All emulsifiers change the intestinal microbiota diversity and induce gut microbiota dysbiosis. Lecithin, sucrose fatty acid esters, and CMC do not impact mucus–bacterial interactions, whereas MDG tends to cause bacterial encroachment into the inner mucus layer and enhance inflammation potential by raising circulating lipopolysaccharide. Our findings demonstrate the safety concerns associated with using dietary emulsifiers, suggesting that they could lead to metabolic syndromes.

Gut microbiota inhabiting the gastrointestinal tract are essential in health and disease. A balanced gut microbiome is functionally beneficial to the body. In contrast, gut microbiota dysbiosis can lead to various illnesses, including cardiovascular disease, obesity, metabolic syndrome, and inflammatory bowel disease, via a metaorganism–pathogenesis pathway involving the gut microbiota, its metabolites, and the host[1–3]. A thick layer of mucus over the intestinal epithelium prevents translocation of gut micro-biota in the body. Colonic bacteria colonizing the outer mucus layer can degrade and utilize mucus glycans as an energy source[4]. Specific mucin-degrading microbiomes have enzyme glycosyl hydrolases that digest specific glycan linkages[5]. A previous study has demonstrated that the intestinal microbiota can influence the properties of the colonic mucus layer[6]. Certain microbes possess various carbohydrate utilization gene clusters that allow them to degrade and metabolize specific glycans in the intestinal mucus layer[5]. Decomposition of the mucus layer leads to gut microbiota encroachment resulting in infection and inflammation. A dysfunctional mucus layer has been found in both murine and human colitis[7]. The change in gut microbiome composition directly influences the changes in the mucus layer. Additionally, our dietary habits, such as intake of a high-fat diet, food additives, and prebiotics, directly impact changes in the mucus, which is

[1]Institute of Food Science and Technology, National Taiwan University, Taipei, Taiwan, ROC. [2]Center for Food and Biomolecules, National Taiwan University, Taipei, Taiwan, ROC. [3]Department of Internal Medicine, College of Medicine, National Taiwan University, Taipei, Taiwan, ROC. [4]Department of Medical Research, National Taiwan University Hospital, Taipei, Taiwan, ROC. [5]Department of Internal Medicine, National Taiwan University Hospital, Taipei, Taiwan, ROC. [6]Bachelor Program of Biotechnology and Food Nutrition, National Taiwan University, Taipei, Taiwan, ROC. [7]Biodiversity Research Center, Academia Sinica, Taipei, Taiwan, ROC. [8]Department of Anatomy, Tzu Chi University, Hualien, Taiwan, ROC. [9]School of Medicine, College of Medicine, National Sun Yat-sen University, Kaohsiung, Taiwan, ROC. [10]National Laboratory Animal Center, National Applied Research Laboratories, Taipei, Taiwan, ROC. [11]Division of Gastroenterology, Perelman School of Medicine, University of Pennsylvania, Philadelphia, PA, USA. [12]Department of Food Science, Rutgers University, New Brunswick, NJ, USA. [13]National Center for Food Safety Education and Research, National Taiwan University, Taipei, Taiwan, ROC. [14]These authors contributed equally: Suraphan Panyod, Wei-Kai Wu. ✉e-mail: mingshiang@ntu.edu.tw; lysheen@ntu.edu.tw

associated with the development of several diseases[8]. Hence, particular food and food additives can disrupt mucus–bacterial interactions and potentially promote gut inflammation-related diseases.

Food additives, such as instant flavoring agents, preservatives, and emulsifiers, have been used in the food industry to extend the shelf life and improve the appearance, taste, and texture of food products. Emulsifiers might be implicated in the pathogenesis of several diseases, including inflammatory bowel disease and metabolic syndrome. Food additive emulsifiers can reduce the interfacial tension between the oil and water phases and incorporate physical force to form a stable emulsion[9]. Carboxymethylcellulose (CMC) and polysorbate 80 (P80) induce intestinal inflammation and metabolic syndrome by thinning of the intestinal mucus layer and altering gut microbiota composition, increasing the gut epithelial permeability and lipopolysaccharide (LPS) levels[10]. Moreover, both P80 and CMC can modify the microbiota and elevate pro-inflammatory potential in the mucosal simulator of the human intestinal microbial ecosystem (M-SHIME), and transplantation of emulsifier-treated M-SHIME suspensions to germ-free mice can induce low-grade inflammation-associated phenotypes and metabolic disease[11]. Another study found that CMC intervention in healthy participants for ~2 weeks can increase postprandial abdominal discomfort and lower gut microbiota diversity, decrease fecal short-chain fatty acids and free amino acids, and enhance microbiota encroachment into the inner mucus layer[12]. These studies raise safety concerns regarding the use of emulsifiers, which may adversely affect health and lead to chronic diseases.

The classification of emulsifiers as food additives differs among regulatory bodies and across countries. Numerous food additives are recognized as emulsifiers by the Joint Food and Agriculture Organization/World Health Organization Expert Committee on Food Additives (JECFA), Codex Alimentarius (Codex), UK Food Standards Agency, and US Food and Drug Administration. However, not all emulsifiers are acknowledged by every organization (Supplementary Fig. 1a, b). The discrepancies among emulsifier classifications create a challenge for international translation of emulsifier research resulting in differing definitions of a low-emulsifier diet across countries[9]. Presently, dietary exposure evaluations for emulsifiers that are commonly used in the population are lacking. A recent report has attempted to estimate the dietary exposure to seven emulsifiers, namely mono- and diglycerides (MDGs), lecithin, CMC, sucrose esters, polysorbate 80 (P80), stearoyl lactylates (sodium stearoyl lactylate (SSL) and calcium stearoyl lactylate (CSL)), and polyglycerol polyricinoleate (PGPR), during two time periods (1999–2002, 2003–2010) (Supplementary Fig. 1c). The exposure levels to emulsifiers remained constant during the period. MDGs, lecithin, and CMC had the highest mean and 90th percentile exposures; nevertheless, JECFA has not published any reports on acceptable daily intake (ADI) values for these three emulsifiers. Conversely, sucrose esters, P80, stearoyl lactylates, and PGPR exhibited lower exposure levels than the aforementioned emulsifiers. The exposure estimates to emulsifiers are in line with the market value data (Supplementary Fig. 1d)[13].

Traditionally, the safety evaluation and approval of food additives were based on toxicological evidence following the Generally Recognized As Safe (GRAS) guidelines. However, with recent discoveries of the significant role of gut microbiota in health and diseases, researchers have shifted their focus to studying the adverse health effects of food additives via the metaorganism–pathogenesis pathway and gut microbiota. Thus, it is necessary to explore the safety of food additives concerning the interactions between gut microbiota and the host to fill in the gaps in current literature and support regulations governing food production. Currently, the knowledge regarding the effect of emulsifiers on the development of metabolic disease is limited. This study aimed to reassess the safety of dietary emulsifiers by examining their impact on gut microbiota and their relation with obesity and metabolic diseases. This study investigated the impact of lecithin, sucrose fatty acid esters, CMC, and MDGs on the development of obesity and metabolic disease through gut microbiota and host interactions, including gut microbiota dysbiosis, changes in the mucus layer, intestinal permeability, and the translocation of gut-derived LPS into the circulatory system. The selection of these emulsifiers was due to the estimated high dietary exposure of humans to them[13]. Moreover, at the beginning of the study, it became evident that MDGs could not be dissolved in water, unlike the remaining emulsifiers. Consequently, we categorized the selected emulsifiers into two sets of experiments to better accommodate their distinct properties and experimental requirements.

## Results

### Intake of dietary emulsifiers sucrose fatty acid esters, and CMC adversely affected the obesogenic and metabolic biomarkers and induced insulin resistance in both in vivo and in vitro models

We investigated the effect of common hydrophilic dietary emulsifiers on obesogenic, metabolic, gut microbiota, and gastrointestinal changes (Fig. 1a). Common hydrophilic dietary emulsifiers used for investigation in this study included lecithin, sucrose fatty acid esters, and CMC. Fifteen-week-old male C57BL/6JNarl mice were fed a regular chow diet with or without emulsifiers supplemented in water for 17 weeks. All emulsifiers were administered via drinking water. Mice in each experimental group had unrestricted access to both food and water. The dosage design for lecithin and sucrose fatty acid esters followed the report on estimated dietary exposure in humans[13]. The human dosage of emulsifiers was translated to equivalent mouse doses. The doses of lecithin and sucrose fatty acid esters were 10 times the respective daily exposure levels in humans, that is 7523.3 and 1110 mg/kg bw/day, respectively. CMC (1% in drinking water) dosage was set according to a previous study and used as a positive control[10]. After ingestion of the dietary emulsifier for 17 weeks, the actual intake for lecithin was nearly double the theoretical intake. In contrast, the actual intake for sucrose fatty acid esters closely matched the theoretical intake (Supplementary Fig. 2). The group with emulsifier supplementation showed a higher weight than the control group (Fig. 1b). At week 17, the CMC-treated group had increased weight gain compared to the control group ($P = 0.0322$), while the group treated with sucrose fatty acid esters showed only a tendency to gain weight ($P = 0.1563$) (Fig. 1c). Similarly, the CMC-fed group revealed increased relative fat mass but decreased lean mass. (Fig. 1d, e). Moreover, the CMC group showed a lower serum total cholesterol and triglyceride level ($P = 0.0034$ and $P = 0.0006$) than the control group (Fig. 1f, g), suggesting that CMC may interfere with lipid absorption and metabolism. The lecithin, sucrose fatty acid esters and CMC did not cause any changes in serum aspartate aminotransferase (AST) and alanine aminotransferase (ALT) levels, fatty liver, and histopathological characteristics in the liver section (Supplementary Fig. 3). Oral glucose tolerance test (OGTT) revealed no notable variation in the area under the curve (AUC) of OGTT between the emulsifier-fed group and the control group (Fig. 1h, i). Sucrose fatty acid ester- and CMC-fed groups showed a significant increase in fasting serum glucose ($P = 0.0029$ and $P = 0.0095$, respectively) and insulin levels ($P = 0.0027$ and $P < 0.0001$, respectively) (Fig. 1j, k). Homeostatic model assessment for insulin resistance (HOMA-IR) revealed that sucrose fatty acid esters and CMC increased the HOMA-IR index (Fig. 1l), suggesting that the intake of the emulsifiers sucrose fatty acid esters and CMC dysregulated glucose and insulin homeostasis. Lecithin supplementation in mice tended to increase serum glucose and insulin levels and the HOMA-IR index; however, no significant difference was observed when compared to the control group.

Given the observed insulin resistance (IR) in the emulsifier-fed mice, we proceeded to investigate whether these emulsifiers directly influence glucose homeostasis by using insulin resistance in vitro model with 3T3-L1 adipocytes (Fig. 1m). The doses of emulsifiers administered to mice were extrapolated to doses for the IR cellular model (at ratios of 1:100 and 1:1000 for each emulsifier) (Supplementary Fig. 4). Lecithin and sucrose fatty acid ester significantly reduced 2-deoxyglucose uptake at 30 min ($P = 0.0010$ and $P < 0.0001$, respectively), suggesting insulin resistance induced by these emulsifiers. At 60 min, lecithin, sucrose fatty acid ester, and carboxymethylcellulose (CMC) exhibited reductions in 2-deoxyglucose uptake ($P < 0.0001$) (Fig. 1n, o). However, the effect size observed with CMC was relatively small compared to the other two emulsifiers. Hence, it may be

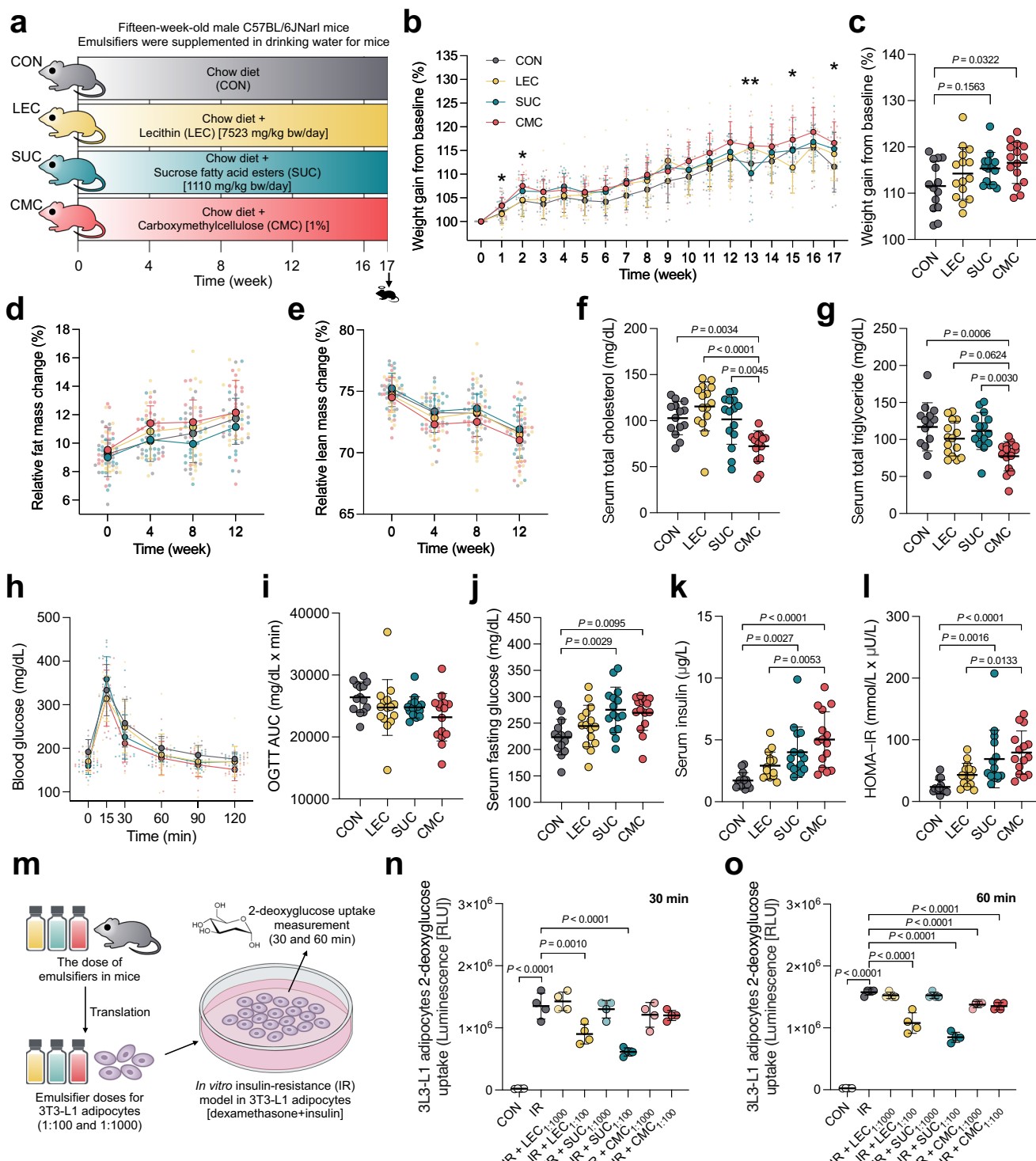

**Fig. 1 | Dietary emulsifiers, including sucrose fatty acid esters and CMC, had an adverse effect on obesogenic and metabolic biomarkers, inducing insulin resistance in vivo and in vitro. a** Animal experimental design, (**b**) changes in weight gain, (**c**) changes in weight gain at 17th week, (**d**) changes in relative fat mass, (**e**) changes in relative lean mass, (**f**) serum total cholesterol levels, (**g**) total triglyceride levels, (**h**) oral glucose tolerance test (OGTT) curve, (**i**) area under the curve (AUC) of OGTT, (**j**) serum fasting glucose levels, (**k**) serum insulin levels, and (**l**) homeostatic model assessment for insulin resistance (HOMA-IR), (**m**) experimental design of in vitro cellular insulin-resistance (IR) model in 3T3-L1 adipocytes, (**n**) 3T3-L1 adipocytes 2-deoxyglucose uptake at 30 min, and (**o**) 60 min. Mice were supplemented with or without different emulsifiers in drinking water for 17 weeks. Dot plots are expressed as the mean ± s.d. ($n$ = 14–15 per group). Insulin resistance (IR) was induced in 3T3-L1 adipocytes using dexamethasone for 72 h. Emulsifiers, including LEC at concentrations of 0.2 mg/mL (1:1000) and 2 mg/mL (1:100), SUC at concentrations of 0.03 mg/mL (1:1000) and 0.3 mg/mL (1:100), and CMC at concentrations of 0.1 mg/mL (1:1000) and 1 mg/mL (1:100), were also introduced into the experimental setup. Statistical analyses were performed using one-way ANOVA with Tukey's range test for comparisons shown as exact $P$ values. *, $P < 0.05$ and **, $P < 0.01$ (CON v.s. CMC group). The 2-deoxyglucose uptake data were analyzed using one-way ANOVA with Dunnett's multiple comparisons test, comparing against the IR group. CON control group, LEC lecithin group, SUC sucrose fatty acid esters group, CMC carboxymethylcellulose group. Illustrations in (**a**, **m**) were created with Keynote.

inferred that lecithin and sucrose fatty acid ester directly contribute to IR whereas IR induced by CMC may be related to the weight gain.

## Lecithin, sucrose fatty acid esters, and CMC transformed the gut microbiota α- and β-diversity indices

Because gut microbiota plays an important role in the progression of obesogenic and metabolic diseases, we analyzed the cecal microbiota composition using the V4 16S rRNA gene sequencing to elucidate the effect of dietary emulsifiers on gut microbiota. The QIIME2 platform was used to process the raw sequence for generating the amplicon sequence variance (ASV) table and subsequently aligned against the SILVA database (version 138) to obtain the taxonomic classification of each ASVs. ASVs table comprised 671 ASVs classifying into 122 species and 86 genera. Lecithin induced α-diversity indices similar to those of the control. The sucrose fatty acid esters group showed a significant elevation in the observed species ($P = 0.0372$), suggesting that this emulsifier increased the abundance of the particular gut microbiome, and the microbial community in the cecum exhibited low evenness (Fig. 2a–c). CMC significantly elevated the Shannon and Simpson diversity indices ($P = 0.0229$, P = 0.0366, respectively), suggesting CMC could make the diversity of gut microbiota more consistent.

We further computed β-diversity and performed principal coordinate analysis (PCoA) based on the Bray–Curtis distance (Fig. 2d). The data indicated that lecithin, sucrose fatty acid esters, and CMC affected the gut microbiota shift. PCoA displayed significant separation among all experimental groups (analysis of similarity [ANOSIM]: R = 0.8205, $P < 0.001$). The distance of the centroid of the lecithin group to the control group was shortest, indicating that lecithin did not strongly affect the cecal microbiome. In general, sucrose fatty acid esters and CMC have a more substantial effect on gut microbiome shift. The sucrose fatty acid esters group displayed a distinct separation from the control group counting on the X-axis (PCoA1; 27.32%), whereas the lecithin and CMC groups were relatively similar to the control. However, the CMC group showed a distinct partition counting on the Y-axis (PCoA2; 20.9%). We also added details of the insulin resistance-related biomarker, including HOMA-IR, blood glucose, and insulin level, in the PCoA plot. The increased circle size for each mouse indicated a higher HOMA-IR value, and the higher intensity of color represented a higher serum glucose or insulin levels. All emulsifier-fed groups showed increased circle sizes and color intensities, suggesting that insulin resistance-related parameters may be correlated with the gut microbiota shift. We further estimated the relationship between each genus and the dimensional space of each mouse's gut microbiome. Vectors in the PCoA plot symbolized an associated genus ($P < 0.01$), and its length presented the strength of the relationship. The control group associated with the genera *Olsenella* is enriched in lean mass[14]. The lecithin group exhibited a relationship with *Faecalibaculum*, *Enterorhabdus*, and *Muribaculum*. The sucrose fatty acid esters group correlated with *[Eubacterium] xylanophilum* group, *Clostridium sensu stricto 1*, and *Akkermansia*, whereas the CMC group correlated with *Alloprevotella*, *Acetatifactor*, *Muribaculaceae*, *Clostridia* vadinBB60 group, *Blautia*, and UCG−010. Collectively, lecithin, sucrose fatty acid esters, and CMC emulsifier feeding modified the gut microbiota in terms of both the α- and β-diversity indices, and each emulsifier was associated with its particular genus and insulin resistance biomarkers.

## Lecithin, sucrose fatty acid esters, and CMC adversely affected gut microbiota

There were 35 significantly different genera among the experimental group. The hierarchical clustering of the genus is classified into three primary clusters characterizing the differences between the emulsifier-fed and control groups. The results indicate that three kinds of emulsifiers differentiated the cecal microbiota at the genus level (Fig. 2e). The head of the heatmap displays the experimental group and percent of body weight change, insulin, and fasting glucose level, which demonstrated raised values in the three

emulsifier-fed groups. Compared with the control group, the lecithin, sucrose fatty acid esters, and CMC groups had 18 (9↑/9↓), 19 (7↑/12↓), and 19 (9↑/10↓) significantly different genera ($P < 0.05$), respectively. The lecithin group showed increased abundances in disease-related bacteria, including *Streptococcus*, *[Eubacterium] coprostanoligenes* group, *Enterobacter*, *Lachnoclostridium*, *Desulfovibrio*, and *[Eubacterium] xylanophilum* group, as well as other genera, such as the *Bifidobacterium*, *Lactobacillus*, and *Candidatus Arthromitus*. Lecithin also reduced the abundance of possible beneficial bacteria, including *Oscillibacter*, *Parasutterella*, *Dubosiella*, and *Turicibacter*, and other genera, such as the *Prevotellaceae* UCG −001, *Colidextribacter*, *Clostridia* vadinBB60 group, *Coriobacteriaceae* UCG−002, and *Tyzzerella*. The sucrose fatty acid ester-fed group was enriched in disease-related microbiome, including *Clostridium sensu stricto 1*, *Lachnospiraceae* UCG−006, and *[Eubacterium] xylanophilum* group, and other genera—the *[Eubacterium] ruminantium* group, *Akkermansia*, *Lactobacillus*, and UCG−010. Sucrose fatty acid esters also depleted possible beneficial bacteria such as *Muribaculaceae*, *Oscillibacter*, *Faecalibaculum*, *Parasutterella*, *Olsenella*, and other genera—the *Alloprevotella*, *Acetatifactor*, *Prevotellaceae* UCG−001, *Colidextribacter*, *Clostridia* vadinBB60 group, *Enterorhabdus*, and *Tyzzerella*. In the CMC group, there was an elevated relative abundance of several disease-associated bacteria, including *Blautia*, *Staphylococcus*, and *[Eubacterium] coprostanoligenes* group, and other genera, such as the *Muribaculaceae*, *Incertae Sedis*, *Clostridia* vadinBB60 group, *Enteractinococcus*, *Candidatus Arthromitus*, and UCG −010. The abundance of several beneficial microbes also lessened in the CMC-treated group, such as *Muribaculum*, *Faecalibaculum*, *Parasutterella*, *Dubosiella*, *Turicibacter*, *Prevotellaceae* UCG−001, *Coriobacteriaceae* UCG −002, *Enterorhabdus*, *Tyzzerella*, and *[Eubacterium] siraeum* group. These data suggested that dietary emulsifiers lecithin, sucrose fatty acid esters, and CMC adversely affected gut microbiota homeostasis by increasing disease-associated microbiomes but depleting particular beneficial microbiomes.

We additionally analyzed Spearman's correlation to evaluate the association of the meaningful genus with obesogenic and metabolic biomarkers based on the heatmap. Changes in body weight positively correlated with two genera: UCG−010 and the disease-associated genera *Blautia*. Fasting blood glucose positively correlated to *Blautia*, *Incertae Sedis*, and the *Clostridia* vadinBB60 group. Insulin and HOMA-IR were positively associated with a mucin degrading bacteria—*Akkermansia*. We observed that several beneficial bacteria, including *Muribaculum*, *Faecalibaculum*, *Parasutterella*, *Olsenella*, and *Dubosiella*, were negatively correlated with changes in body weight, fasting glucose levels, insulin levels, or HOMA-IR index. The heatmap and correlation results collectively demonstrated that lecithin, sucrose fatty acid esters, and CMC adversely affected gut microbiota and may cause microbiota dysbiosis.

## Lecithin, sucrose fatty acid esters, and CMC did not disrupt mucus–bacterial interactions or promote diseases associated with gut inflammation

The shortening of the colon length is considered to a biomarker of colitis. The colon length in the hydrophilic emulsifiers-fed group was not significantly different from that of the control group (Fig. 2f). We used periodic acid-Schiff (PAS)-stained colon sections to obtain the colonic epithelial damage histological score. The results showed that the intestinal epithelial cells in both the emulsifier-treated and control groups were structurally intact, the number of goblet cells was normal, and the lamina propria and muscular mucosae were not infiltrated by immune cells, suggesting that the administered lecithin, sucrose fatty acid esters, and CMC did not induce colitis (Supplementary Fig. 5). We further investigated whether these dietary emulsifiers affected the intestinal mucus layer changes and localization of bacteria using confocal microscopy. Lecithin, sucrose fatty acid esters, and CMC intervention did not result in the thinning of the mucus layer and shortening of the distance of bacteria to intestinal epithelial cells (IECs), indicating there was no invasion of intestinal bacteria to the inner mucus layer, and these emulsifiers did not facilitate diseases associated with gut inflammation (Figs. 2g and 3h). Thus, the intake of hydrophilic dietary

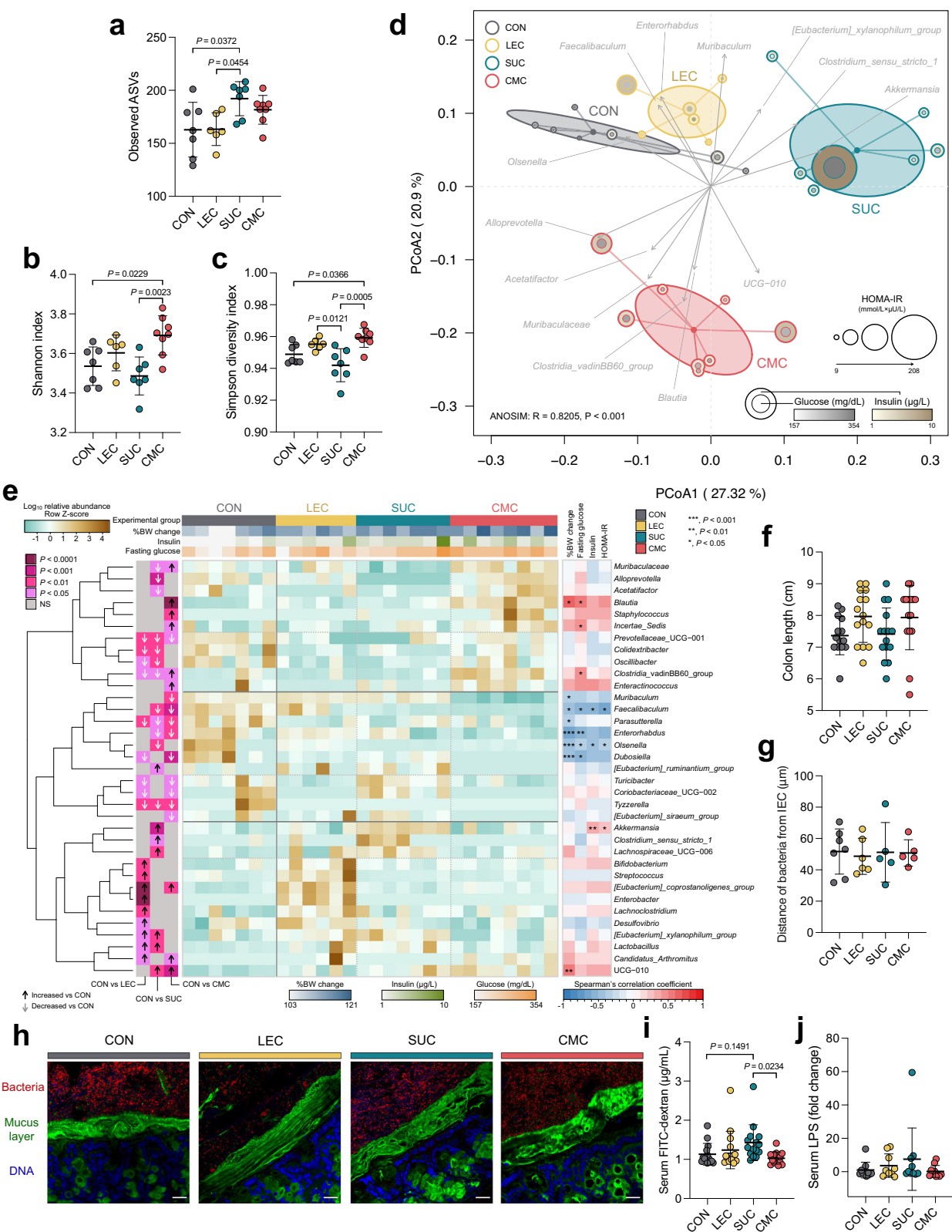

emulsifier did not disrupt mucus–bacterial interactions. Next, we examined the effect of emulsifiers on intestinal permeability using oral gavage of the fluorescein isothiocyanate (FITC)-dextran method. The sucrose fatty acid esters group showed increased intestinal permeability ($P = 0.1491$) compared to the control group, unlike lecithin and CMC (Fig. 2i). LPS produced

by gut microbiota can translocate to the circulation system through the portal vein and result in low-grade systemic inflammation. Our data showed that the serum LPS level and intestinal permeability were consistent in mice fed with emulsifiers (Fig. 2j); however, the LPS level was not statistically different.

**Fig. 2 | Dietary emulsifiers lecithin, sucrose fatty acid esters, and CMC transformed gut microbiota diversity indices, adversely affecting the gut microbiota, but did not disrupt mucus–bacterial interactions or promote gut inflammation-associated diseases. a** Observed amplicon sequence variances (ASVs), (**b**) Shannon index, (**c**) Simpson diversity index, (**d**) principal coordinate analysis (PCoA) plot based on Bray–Curtis dissimilarity with gut microbiome-associated vector (envfit {vegan}), (**e**) heatmap of the relative abundances of cecal microbiota with significant differences determined using the Kruskal–Wallis test ($P < 0.05$) and with Spearman's correlation coefficient analysis between gut microbiome genera and obesogenic and metabolic biomarkers, (**f**) colon length, (**g**) distances of closest bacteria to intestinal epithelial cells (IECs), (**h**) representative image of localization of bacteria using fluorescent in situ hybridization, (**i**) serum FITC-dextran concentration, and

(**j**) serum lipopolysaccharide (LPS) levels. Mice were supplemented with or without different emulsifiers in drinking water for 17 weeks. Dot plots represent the mean ± s.d. ($n = 5–8$/group). Statistical analyses were performed using one-way ANOVA with the Tukey's range test for comparisons shown as exact $P$ values. Analysis of similarity (ANOSIM) was used to analyze the heterogeneity of the cecal microbiota among the groups in PCoA. Vectors in the PCoA plot show a significant genus ($P < 0.01$), and its length indicates the strength of the correlation. Pairwise statistical analyses were performed using an unpaired Wilcoxon signed-rank test shown as heatmap (CON vs. LEC; CON vs. SUC; and CON vs. CMC). Confocal microscopy analysis of microbiota localization: mucus layer, green; bacteria, red; and DNA, blue. Scale bar, 25 μm. CON control group, LEC lecithin group, SUC sucrose fatty acid esters group, CMC, carboxymethylcellulose group.

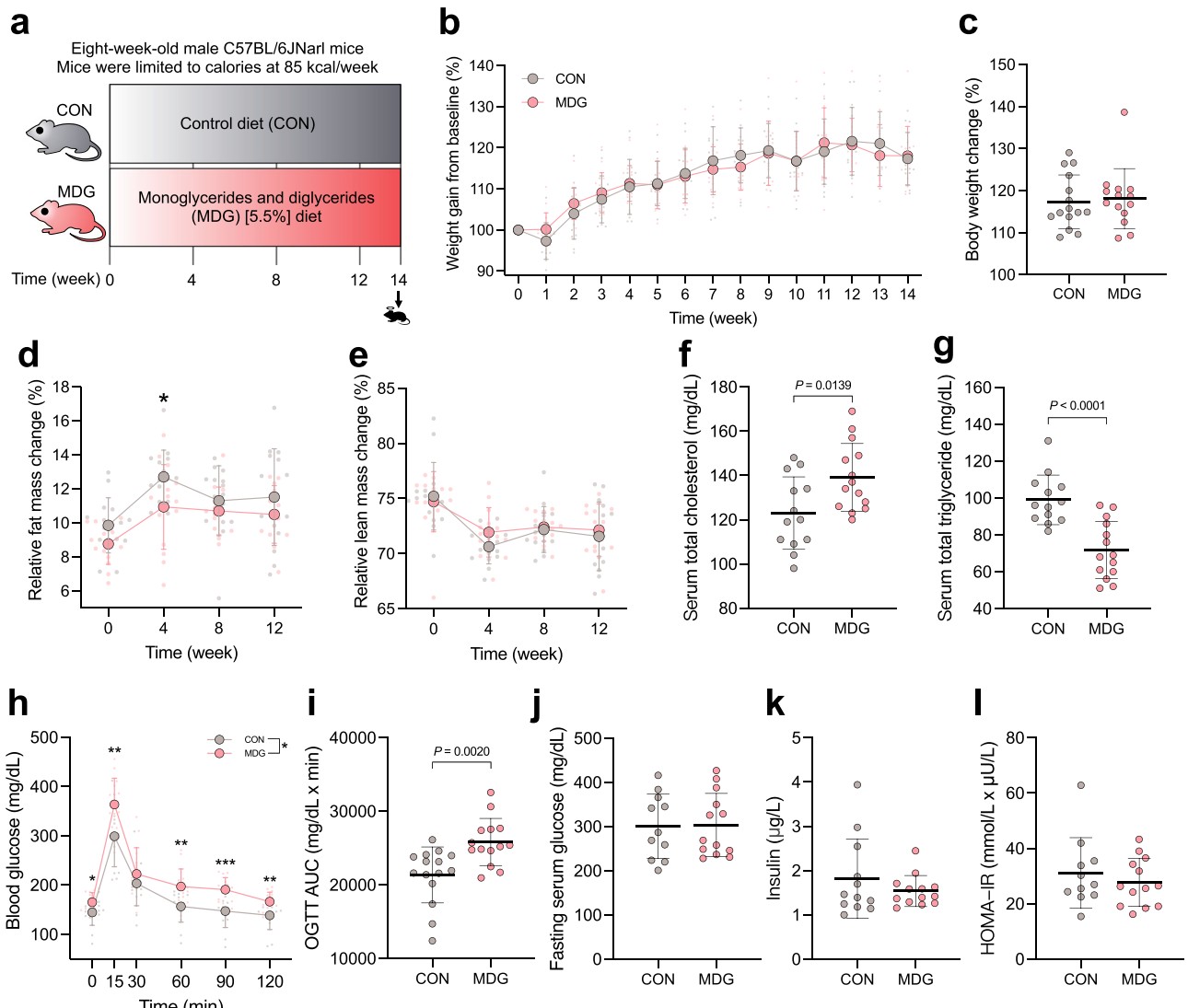

**Fig. 3 | Dietary emulsifier mono- and diglycerides impaired circulating lipid and glucose metabolism. a** Animal experimental design, (**b**) changes in weight gain, (**c**) changes in weight gain at the 14th week, (**d**) changes in relative fat mass, (**e**) changes in relative lean mass, (**f**) serum total cholesterol levels, (**g**) total triglyceride levels, (**h**) oral glucose tolerance test (OGTT) curve, (**i**) area under the curve (AUC) of OGTT, (**j**) serum fasting glucose levels, (**k**) serum insulin levels, and (**l**) homeostatic model

assessment for insulin resistance (HOMA-IR). Mice were fed control or MDG diet for 14 weeks. Dot plots are expressed as the mean ± s.d. ($n = 14–15$ per group). Statistical analyses were performed using an unpaired two-tailed Student's $t$-test for comparisons (CON vs. MDG) and shown as exact $P$ values or symbols (*, $P < 0.05$). CON control group, MDG monoglycerides and diglycerides group. Illustrations in (**a**) were created with Keynote.

## Mono- and diglycerides (MDGs) impaired circulating lipid and glucose metabolism

We investigated the common emulsifier MDG (Fig. 3a). Eight-week-old male C57BL/6JNarl mice were fed with a diet containing MDG 5.5% diet.

The dosage of MDG used in this experiment was based on that of diacylglycerol used in a prior study, showing no signs of systemic toxicity[15]. As both soybean oil and MDG are fats that provide energy to the body, and the control diet contained triacylglycerol mainly derived from soybean oil, we

substituted soybean oil with MDG in the diet fed to the MDG group. The mice were fed with control or MDG diet for 14 weeks. To minimize the effect of different amounts of dietary intake in each mouse, mice were limited to ~85 kcal/week[16]. After intervention with MDG for 14 weeks, no significant differences with respect to changes in body weight at 14 weeks was observed (Fig. 3b, c). However, the relative fat mass of the MDG group during feeding was slightly lower than that in the control group, but no significant difference in relative lean mass change was observed (Fig. 3d, e). The MDG intake interfered with blood lipid metabolism by elevating the total serum cholesterol levels ($P = 0.0139$) and lowering the total triglyceride levels ($P < 0.0001$), compared to the control group (Fig. 3f, g). However, MDG did not affect serum AST and ALT levels (Supplementary Fig. 6). OGTT revealed that MDG significantly increased the blood glucose levels. The calculated AUC of OGTT was significantly higher for the MDG group than for the control ($P < 0.0020$) (Fig. 3h, i). However, no differences in the fasting serum glucose levels, serum insulin levels, and HOMA-IR index were observed between the MDG and control groups (Fig. 3j–l). These data suggested that MDG intake impaired glucose and lipid metabolism.

### MDG reduced the evenness of the gut microbiota community and modified the gut microbiome β-diversity

According to the QIIME2 pipeline, the V4 16S rRNA gene sequences generated 460 ASVs and were assigned to 110 species and 80 genera. MDG did not impact the observed ASVs or Shannon diversity index but reduced the microbiota community evenesss, the Simpson's diversity index ($P = 0.0285$) (Fig. 4a–c). These data suggest that MDG allow or inhibit the growth of a particular microbiome, resulting in the unevenness of the gut microbiota community. The Bray−Curtis distance-based PCoA demonstrated that MDG simulated the gut microbiota shift (Fig. 4d) with significant separation (ANOSIM: R = 0.9378, $P < 0.001$). The MDG group displayed a distinguishable separation from the control group, on the X-axis (PCoA1; 47.88%). Y-axis (PCoA2; 15.96%) showed the effect of the intra-group microbiota shift. The gut microbiota of each mouse in the PCoA plot of the MDG group exhibited a shorter distance toward its group centroid, suggesting MDG firmly adjusted gut microbiota composition with the lower within-group variation. In contrast, the control group displayed a higher inter-group variation. The PCoA plot also showed the lipid and glucose metabolism biomarkers, including body weight change, AUC of OGTT, and total serum cholesterol. The larger circle indicated a higher body weight increment, and a more intense color showed a more elevated AUC of OGTT and cholesterol level. MDG tended to cause a reduction in the circle's size but increased the color intensity, suggesting that the body weight, lipid, and glucose metabolism biomarkers may be associated with changes in the gut microbiota. According to the vector orientation with statistical significance set at $P < 0.05$, the MDG group was associated with the genera *Lactococcus, Enterorhabdus*, and *uncultured*. The control group was associated with *Collinsella, Alistipes, Parabacteroides, Alloprevotella*, ASF356, *Lachnospiraceae* NK4A136 group, *Clostridia* UCG−014, *Muribaculaceae, Lachnospiraceae* UCG−006, *Coriobacteriaceae* UCG−002, and *Akkermansia*. Therefore, MDG altered the cecal microbiota's α-diversity evenness and β-diversity index, which is associated with impaired blood lipid and glucose metabolic biomarkers.

### Dietary emulsifier MDG causes gut microbiota dysbiosis

We further performed the statistical analysis of gut microbiota at the genus level based on the Wilcoxon signed-rank test for comparisons ($P < 0.05$) of the effect of MDG on gut microbiota (Fig. 4e). Sixteen genera were significantly different from those in the control group. The hierarchical clustering found two main clusters representing the dissimilarities between the MDG-fed and control groups. Eleven genera were significantly reduced, whereas five genera were enriched in the MDG-fed group. MDG intake decreased possible beneficial bacteria, including *Muribaculaceae, Parabacteroides, Lachnospiraceae* NK4A136 group, *Akkermansia, Collinsella*, and *Coriobacteriaceae* UCG−002, and few other genera, such as *Alistipes, Alloprevotella*, ASF356, A2, and *Lachnospiraceae* UCG−006. Moreover, the

MDG group increased several disease-related microbiomes such as *Enterorhabdus, Jeotgalicoccus*, and *Atopostipes*, and few other genera, such as *uncultured* and *Lactococcus*. Hence, consumption of MDG may result in gut microbiota dysbiosis.

Seven genera (*Alistipes, Parabacteroides, Lachnospiraceae* NK4A136 group, *Alloprevotella*, ASF356, *Lachnospiraceae* UCG−006, and *Collinsella*) were positively correlated to changes in body weight, whereas *Enterorhabdus* exhibited a negative correlation. AUC of OGTT was negatively associated with *Alloprevotella*, ASF356, *Akkermansia*, and *Lachnospiraceae* UCG−006, and *Coriobacteriaceae* UCG−002 demonstrated a negative correlation with total blood cholesterol. The heatmap and correlation results collectively showed that the common emulsifier MDG adversely affected gut microbiota composition and may be associated with changes in metabolic biomarkers.

### MDG slightly decreased the distance of bacteria to epithelial cells and enhanced inflammation potential by increasing circulating LPS

Next, we examined how MDG affected the physiological change in the colon, intestinal permeability, and translocation of the bacterial product. The colon length of mice in the MDG-fed group was not significantly different from that in the control group (Fig. 4f). The PAS-stained colon sections and colonic epithelial damage histological score demonstrated that MDG did not induce colitis (Supplementary Fig. 7). Furthermore, although the distance of bacteria to IECs was shorter in mice fed the MDG diet than in those fed the control diet, this difference was not statistically significant ($P = 0.1780$) (Fig. 4g, h), suggesting that the MDG-altered gut microbiome may enter the inner layer of the mucus. The reduced distance between bacteria and epithelial cells may promote gut inflammation-associated disease. Additionally, MDG did not affect FITC-dextran-based intestinal permeability (Fig. 4i). However, the serum LPS level was significantly elevated in mice fed MDG (Fig. 4j), suggesting that the MDG-altered microbiota increased LPS production.

### Discussion

This study demonstrated that common dietary emulsifiers could potentially cause metabolic disorders and gut microbiota dysbiosis, and different emulsifiers showed particular effects on health-related biomarkers. Intake of sucrose fatty acid esters and CMC resulted in an adverse effect on obesogenic and metabolic biomarkers and induced hyperglycemia and hyperinsulinemia by increasing weight gain, fasting serum glucose and serum insulin levels, and HOMA-IR index. The CMC group tended to increase the relative fat mass but decreased the lean mass and interfered with lipid absorption and metabolism by lowering serum total cholesterol and triglyceride levels compared to the control group. In contrast, lecithin supplementation did not show significant increases in serum glucose and insulin levels and HOMA-IR index. The in vitro insulin resistance model demonstrated that lecithin and sucrose fatty acid ester significantly reduced 2-deoxyglucose uptake, whereas CMC had a lesser impact, indicating a direct impact on insulin resistance by these emulsifiers. Besides, MDG affected blood lipid and glucose metabolism by increasing serum cholesterol but reducing triglyceride, thereby raising the blood glucose level of the OGTT.

In the human body, enzymatic digestion of sucrose fatty acid esters produces sucrose and fatty acids or fructose, glucose, and fatty acids. Intervention with a high-sucrose or high-fructose diet influences blood glucose metabolism by reducing insulin sensitivity and increasing fasting blood glucose and insulin concentration[17,18]. Therefore, abnormal hyperglycemia and hyperinsulinemia in the sucrose fatty acid esters group are caused by the increased uptake of sugar and fatty acid. Moreover, dietary sweeteners sucralose and saccharin supplementation impair glycemic response linked to the microbiome in healthy adults[19]. According to findings from our in vitro study, lecithin and sucrose fatty acid ester directly enhanced insulin resistance in the 3T3-L1 adipocytes, whereas CMC had a relatively smaller effect. In contrast, our animal study revealed notable

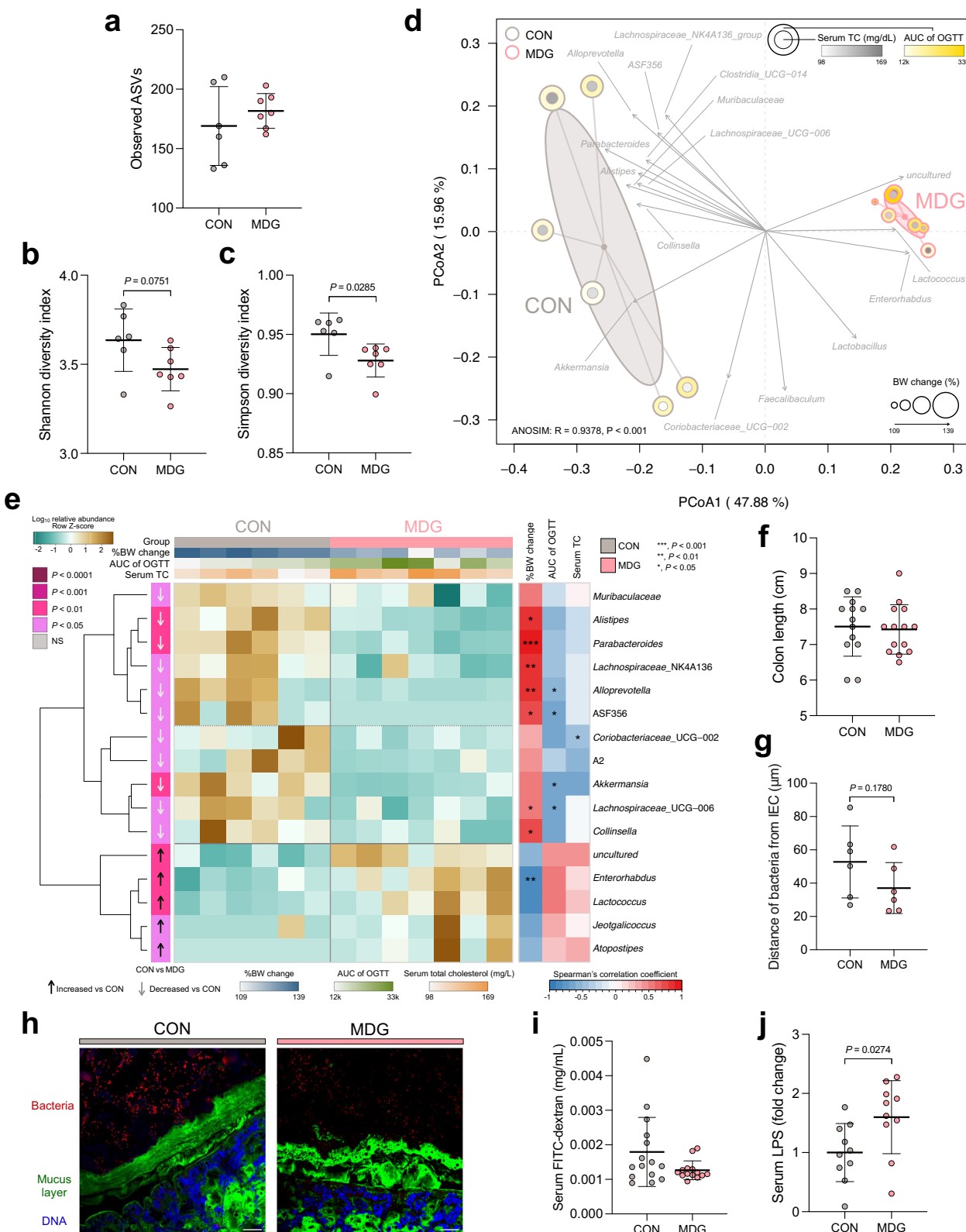

increases in fasting glucose, serum insulin, and HOMA-IR in the CMC group compared to other emulsifiers used in this study. Furthermore, a significant body weight gain was observed in the CMC group. These findings suggest a potential association between the heightened insulin resistance in the CMC group and the observed increase in body weight. CMC and P80 induce metabolic syndrome and intestinal inflammation by thinning the intestinal mucus layer, altering gut microbiota composition, and increasing the gut epithelial permeability and the LPS level[10]. This shift in gut microbiota induced by CMC and P80 leads to low-grade inflammation-associated phenotypes and metabolic disease in the germ-free mice model[11]. CMC can increase postprandial abdominal discomfort, affect gut microbiota shift, lower beneficial fecal metabolite, and enhance bacterial

**Fig. 4 | Dietary emulsifier mono- and diglycerides modified the gut microbiome diversity, causing gut microbiota dysbiosis, decreased the distance of bacteria to epithelial cells, and potentially enhanced inflammation by increasing circulating LPS levels. a** Observed amplicon sequence variances (ASVs), (**b**) Shannon index, (**c**) Simpson diversity index, (**d**) principal coordinate analysis (PCoA) plot based on Bray–Curtis dissimilarity with gut microbiome-associated vector (envfit {vegan}), (**e**) heatmap of the relative abundances of cecal microbiota with significant differences measured using an unpaired Wilcoxon signed-rank test ($P < 0.05$) and with Spearman's correlation analysis between gut microbiota genera and obesogenic and metabolic biomarkers, (**f**) colon length, (**g**) distances of closest bacteria to intestinal epithelial cells (IECs), (**h**) representative image of the localization of bacteria using

fluorescent in situ hybridization, (**i**) serum FITC-dextran concentration, and (**j**) serum lipopolysaccharides (LPS) levels. Mice were fed control or MDG diet for 14 weeks. Dot plots represent the mean ± s.d. ($n = 6$–7/group). Statistical analyses were performed using an unpaired two-tailed Student's $t$-test (CON vs. MDG), and the exact $P$ value is shown. Analysis of similarity (ANOSIM) was used to analyze the heterogeneity of the cecal microbiome among the groups in PCoA. Vectors in the PCoA plot represented a significant genus ($P < 0.05$), and its length indicated the strength of the correlation. Confocal microscopy analysis of microbiota localization: mucus layer, green; bacteria, red; and DNA, blue. Scale bar, 25 μm. CON control group, MDG monoglycerides and diglycerides group.

encroachment into the inner mucus layer in humans[12]. Another study found that CMC and P80 negatively affect physiology and behavior, including anxiety-related and social behaviors, along with a shift in gut microbiota via different mechanisms in males and females[20]. Moreover, CMC and P80 alter gene expression in mouse brains, which potentially impacts stress responses[21]. However, not all dietary emulsifiers have an adverse effect on health. Glycerol monodecanoate, a medium-chain monoacylglycerol, positively impacts the gut microbiota and improves lipid metabolism, insulin sensitivity, and inflammation[22].

Several recent studies demonstrated that food emulsifiers modify the gut microbiota composition and implicate the progression of several chronic diseases, including inflammatory bowel disease and metabolic syndrome. This study found that emulsifiers reshape the gut microbiota α- and β-diversity indices. Lecithin had a lesser influence on gut composition. A previous study reported that lecithin did not significantly influence microbiota in ex vivo in the MiniBioReactor Array model[23]. In contrast, the sucrose fatty acid esters group increased the observed species in gut microbiota but decreased the evenness index. CMC increased the evenness of the gut microbiota community. β-Diversity of these emulsifier-treated groups was associated with an insulin resistance-related biomarker. Moreover, MDG lessened the evenness of the gut microbiota community and shaped the gut microbiome at β-diversity and was associated with impaired blood lipid and glucose metabolic biomarkers.

In general, dietary emulsifiers adversely impacted gut microbiota at the genus level and caused gut microbiota dysbiosis. A previous study examined the effect of 20 emulsifiers on human microbiota shift in an ex vivo model and revealed that most emulsifiers have detrimental consequences on microbiota composition and function[23]. A study investigated the effect of five emulsifiers, namely CMC, P80, soy lecithin, sophorolipids, and rhamnolipids, and revealed that all selectively enriched the abundance of putative pathogens and increased flagellin levels[24]. In this study, the lecithin group showed enrichment of disease-related genera, including *Streptococcus*[25,26], *[Eubacterium] coprostanoligenes* group[27], *Enterobacter*[28,29], *Lachnoclostridium*[30,31], *Desulfovibrio*[32], and *[Eubacterium] xylanophilum* group[33], but reduction of probable health-related bacteria, including *Oscillibacter*[34], *Parasutterella*[35,36], *Dubosiella*[37], and *Turicibacter*[38]. The sucrose fatty acid esters group also showed enrichment of disease-related genera, including *Clostridium sensu stricto 1*[39], *Lachnospiraceae* UCG −006[40], and *[Eubacterium] xylanophilum* group[33], and reduction of possible beneficial genera, such as *Muribaculaceae*[41,42], *Oscillibacter*[34], *Faecalibaculum*[43], *Parasutterella*[35,36], and *Olsenella*[14]. The CMC group showed enrichment of the disease-associated genera *Blautia*[44–46], *Staphylococcus*[47], and *[Eubacterium] coprostanoligenes* group[28], whereas the abundance of several beneficial genera, such as *Muribaculum*[48], *Faecalibaculum*[43], *Parasutterella*[35,36], *Dubosiella*[37], and *Turicibacter*[38], was reduced by this emulsifier.

Administration of the dietary emulsifier MDGs also caused gut microbiota dysbiosis. MDG reduced the abundance of several potentially beneficial bacteria, including *Muribaculaceae*, *Parabacteroides*[49], *Lachnospiraceae* NK4A136 group[50], *Akkermansia*[51], *Collinsella*[52], and *Coriobacteriaceae* UCG−002[53], but boosted the growth of several disease-related

microbiomes, such as *Enterorhabdus*[54], *Jeotgalicoccus*[55], and *Atopostipes*[56]. Comprehensive information on the association of gut microbiota taxa whose abundance changed in groups treated with these emulsifiers with health- and disease-related factors is provided in Supplementary Table 1. It is imperative to emphasize that while the information regarding the association of gut microbiota taxa delineated in Supplementary Table 1 is sourced from existing literature, it is crucial to acknowledge that alternative publications may have identified the same bacterium within disparate contexts. This acknowledgment underscores an inherent limitation in our discussion. In summary, these data suggested that common dietary emulsifiers lecithin, sucrose fatty acid esters, CMC, and MDG demonstrated an adverse effect on gut microbiome homeostasis by elevating disease-associated bacterial genera and reducing the relative abundance of beneficial microbiomes.

Our study uncovered that lecithin, sucrose fatty acid esters, and CMC did not change the colon length, induce colitis, cause thinning of the mucus layer, or shorten the distance between bacterial and IECs. Sucrose fatty acid esters tended to increase intestinal permeability and LPS levels. These data suggested that these emulsifiers did not disrupt mucus–bacterial interactions or facilitate diseases related to gut inflammation. Similarly, MDG did not change the colon length and induce colitis. However, MDG decreased the distance between bacterial and IECs which may promote gut inflammation-associated diseases. Moreover, MDG did not change intestinal permeability but increased serum LPS levels, suggesting that MDG altered the gut microbiota composition and was capable in increasing LPS generation and may result in systemic inflammation. CMC and P80 promote low-grade inflammation, metabolic syndrome, and colitis in mice by inducing microbiota encroachment, altering bacteria composition, and increasing intestinal permeability and LPS levels[10]. Our study demonstrated that CMC induced metabolic disorders and altered bacteria composition, but did not reduce the distances of the closest bacterial cells to IECs. The inconsistency in our findings is possibly attributed to differences in the number of measurements of the closest bacteria to IECs, duration of emulsifier treatment, and gut microbiota composition. Excessive mucin degradation by gut microbiota may cause intestinal disorders allowing luminal antigens to translocate to the intestinal immune system[57]. Mucin-degrading microbes possess glycosyl hydrolases that can digest specific glycan linkages. *Akkermansia glycaniphila* and *muciniphila* have the mucin-degrading gene[5]. Sucrose fatty acid ester supplementation increased *Akkermansia*; however, we did observe a reduction in the distance between bacterial and IECs in sucrose fatty acid ester-fed mice. Beyond the genus *Akkermansia*, 24 genera of bacteria harboring mucin-degrading glycosyl hydrolase gene in various phyla have been reported in the literature. Unfortunately, those genera were not detected in MDG-fed mice, showing a reduced distance between bacterial and IECs.

A high-fat diet is widely recognized as a contributor to insulin resistance[58]. The exact contribution of emulsifiers to insulin homeostasis, and whether the latter is influenced primarily by emulsifier intake or weight gain warrant further investigation. Moreover, we utilized two different forms of emulsifier administration, namely drinking water and dietary supplementation. This approach may have contributed to the observed

differences in outcomes, making it challenging to directly compare the effects among emulsifiers. In future studies aiming to compare the effects of these emulsifiers, both should be administered through dietary supplementation to minimize potential sources of variability. Additionally, conducting a broader comparison including various types of emulsifiers could yield more comprehensive and valuable insights.

Although numerous studies have discovered the negative effect of these food additives, including emulsifiers, the usage of food additives has continued to increase in the food industry. Every country has its own regulations to control the use of food additives. Some nations may follow the GRAS guidelines, established using scientific evidence, to approve one substance as a food additive. Since the discovery of the effects of gut microbiota on health in the past decade, researchers have focused on the effects of food additives on gut microbiota, its function, and its metabolites, which may synergistically cooperate with food additives to adversely affect health, eventually leading to the development of chronic diseases via the metaorganism–pathogenesis pathway. Thus, the safety of food additives with respect to the gut microbiota and host interaction needs to be further investigated to fill gaps in the existing literature and support laws governing food production.

Our findings uncovered that the sucrose fatty acid esters and CMC exhibited the potential to induce obesity and metabolic disorders and resulted in hyperglycemia and hyperinsulinemia. Lecithin and sucrose fatty acid esters directly promote insulin resistance, whereas insulin resistance induced by CMC may be related to weight gain. MDG impaired circulating lipid and glucose metabolism. All emulsifiers remodeled the complexity of gut microbiota composition, increasing the disease-associated microbiome and enhancing gut microbiota dysbiosis. However, lecithin, sucrose fatty acid esters and CMC did not affect mucus–bacterial interactions nor facilitate gut inflammation-associated disease. In contrast, the MDG facilitated bacterial encroachment toward IECs and enhanced inflammation potential by elevating circulating LPS (Fig. 5). Our study provides information regarding the safety concerns associated with the use of dietary emulsifiers, which may help to reevaluate food safety policies and laws governing food production. However, these outcomes necessitate further verification in humans.

## Methods

### Lecithin, sucrose fatty acid esters, and CMC dietary emulsifiers-fed C57BL/6JNarl mouse model

The animals were handled following the guidelines of the Institutional Animal Care and Use Committee of National Taiwan University (approval number: NTU107-EL-00121). Male C57BL/6JNarl mice were purchased from the National Laboratory Animal Center (Taipei, Taiwan). The mice were housed in an animal room with a 12 h light/dark cycle at 23 ± 2 °C. After acclimation, 15-week-old mice were randomly divided into four experimental groups ($n = 3$–4/cage): (i) control, (ii) lecithin (7523.3 mg/kg bw/day, 4.625% in drinking water), (iii) sucrose fatty acid esters (1110 mg/kg bw/day, 0.68% in drinking water), and (iv) CMC (1% in drinking water). CMC was used as a positive control, and its dosage was selected based on a previous study[10]. The dosages of lecithin and sucrose fatty acid esters were selected based on a report on the estimated dietary exposure of humans (lecithin: 61 mg/kg bw/day; sucrose fatty acid esters: 9 mg/kg bw/day)[13] and translated to the corresponding mouse dosages[59]. Emulsifier dosages were calculated based on the average body weight of mice within each experimental group. Emulsifier doses in this experiment were 10 times the respective daily exposure levels in humans. Emulsifiers were purchased from Gemfont Corporation (Taipei, Taiwan): soy lecithin (LASENOR, India; #15682), sucrose fatty acid ester (Mitsubishi Chemical Corporation, Japan; Ryoto Sugar Ester P-1570 (sucrose palmitate)), and sodium carboxymethylcellulose (Nouryon, Finland; #8021514). Mice were fed with a normal chow diet (MFG; Oriental Yeast Co., Ltd., Tokyo, Japan), and all dietary emulsifiers were supplemented in drinking water. Mice in each experimental group were allowed free access to food and water for 17 weeks. Before sacrifice, the mice fasted for 12 h. The mice were subsequently sacrificed using $CO_2$ asphyxiation. Blood was collected via cardiac puncture using a syringe. Organs were collected for subsequent analysis.

### Cell culture, differentiation, insulin-resistance model, and glucose uptake assay

3T3-L1 pre-adipocytes (ATCC CL-173) were cultured and maintained in a pre-adipocyte expansion medium, which consisted of low-glucose (1 g/L) Dulbecco's modified Eagle's medium (DMEM; Invitrogen, USA),

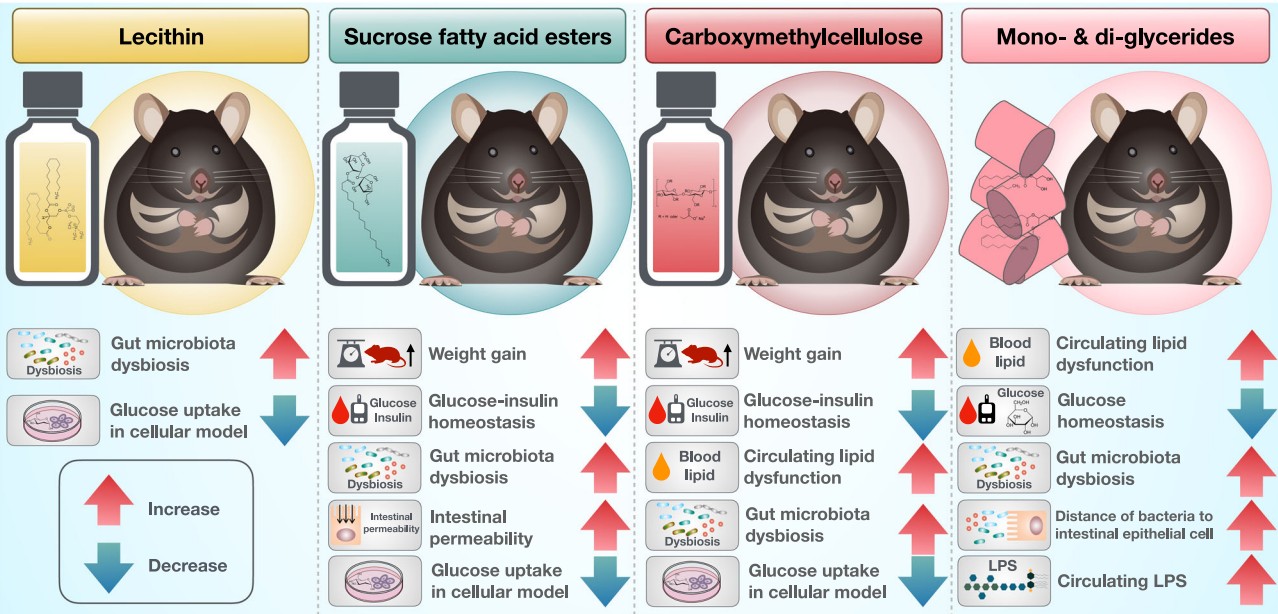

**Fig. 5 | Dietary emulsifiers promote metabolic disorders and induce intestinal microbiota dysbiosis.** The study revealed that dietary emulsifiers lecithin, sucrose fatty acid esters, and carboxymethylcellulose (CMC) were found to disturb glucose-insulin homeostasis, while mono- and diglycerides (MDG) disrupted blood lipid levels and glucose homeostasis. Additionally, these emulsifiers caused imbalances in the gut microbiota. Notably, MDG further exacerbated the increase in gut-derived LPS levels in the bloodstream. These findings provide insights into the risks linked to the consumption of dietary emulsifiers through the lens of gut microbiota, and may prompt a reevaluation of existing food safety policies and regulations governing food production. Illustrations in this figure were created with Keynote.

supplemented with 10% bovine calf serum (BCS) and 1% penicillin-streptomycin solution, at 37°C in a 5% $CO_2$ incubator. The differentiation procedure of 3T3-L1 cells into adipocytes followed a previous publication with modifications[60]. Cells were seeded in a 96-well plate at a density of $1\times10^4$ cells per well and grown for 2 days until reaching 70% confluency. Subsequently, the cells were treated with 1 µM dexamethasone (Sigma-Aldrich, USA, D4902), 1 µg/ml insulin (Sigma-Aldrich, UAS, I9278), and 0.5 mM 3-isobutyl-1-methylxanthine (IBMX; Sigma-Aldrich, USA, 5879) for 72 h in low-glucose DMEM containing 10% fetal bovine serum (FBS) and 1% penicillin-streptomycin solution.

At the endpoint of adipocyte differentiation, adipocytes were washed with phosphate-buffered saline (PBS) and rinsed with an assay medium. The doses of emulsifiers administered to mice were converted to doses appropriate for the insulin resistance (IR) model in mature 3T3-L1 cells. Ratios of 1:100 and 1:1000 were used for each emulsifier, as depicted in Supplementary Fig. 4. For the glucose uptake assay, adipocytes were treated with assay medium supplemented with 10 µg/mL insulin, 1 µM dexamethasone, and emulsifiers (LEC, 0.2 mg/ml and 2 mg/ml; SUC, 0.03 mg/ml and 0.3 mg/ml; CMC, 0.1 mg/ml and 1 mg/ml; all emulsifiers were dissolved with PBS) for 30 and 60 min. Glucose uptake was measured using the Glucose Uptake-Glo Assay (Promega, USA, J1342), following the manufacturer's instructions.

## MDG dietary emulsifiers-fed C57BL/6JNarl mouse model

The mice were housed in an animal room with the previously described conditions. Six-week-old male C57BL/6JNarl mice were employed for the experiment after an adaptation period of 2 weeks. Eight-week-old male C57BL/6JNarl mice were divided into two groups based on their diet: (i) control diet group, and (ii) MDG diet [5.5%] group. The MDG dosage for this experiment followed a previous study that studied the effect of 5.5% diacylglycerol (DAG), exhibiting no signs of systemic toxicity[15]. The control mice were fed with an AIN-93M mature rodent diet. The MDG diet was customized from the AIN-93M mature rodent diet (#D10012M; Research Diets, Inc.) by replacing soybean oil with MDG, and MDG composition was 5.5% in the diet. The nutrition facts of the AIN-93M and MDG diet are shown in Supplementary Table 2. Mice were limited to 85 kcal/week[16], receiving 12.14 kcal/mouse (~3.20 g/mouse) daily. Calorie control was implemented to minimize the differences in dietary intake of the mice for solely observing the effect of MDG. The mice were sacrificed after 14 weeks of the experiment. Before sacrifice, mice fasted for 12 h. After $CO_2$ asphyxiation, the blood was collected by cardiac puncture using a syringe and the organs were collected for subsequent analysis.

## Body composition analysis

Both lean and fat mass body composition was measured using a Minispec LF50 TD-NMR body composition analyzer (Bruker, Billerica, MA, USA). The relative fat or lean mass was calculated as the fat or lean mass to the body weight, respectively.

## Mouse blood biochemistry analysis

The serum was extracted by centrifuging the blood at $1000 \times g$ for 15 min at 4 °C. Serum biochemical biomarkers, including total cholesterol, total triglyceride, high-density lipoprotein (HDL-c), AST, ALT, and glucose, were measured using commercial test strips (Spotchem II reagent strips; Arkray Inc., Kyoto, Japan) in an automatic blood analyzer (Spotchem EZ).

## OGTT

The mice were fasting for 5 h before OGTT experiments. A blood sample was collected from the submandibular vein. Then the blood glucose was analyzed using a glucometer (ACCU-CHEK® Performa, Roche, Basel, Switzerland) at 0, 15, 30, 60, 90, and 120 min after oral gavage with 2 g/kg glucose.

## Intestinal permeability analysis

Intestinal barrier permeability was investigated by administering the intestinal permeability probe FITC-dextran (MW 4,000; Sigma-Aldrich,

46944). After 4 h of fasting, the mice were gavaged with 0.2 mL of FITC-dextran solution (15 mg of FITC-dextran in phosphate-buffered saline). Blood was collected after 3 h. The serum was extracted by centrifuging the blood at $1000 \times g$ for 15 min at 4 °C. Plasma FITC-dextran fluorescent intensity was measured using a fluorometer in black 96-well plates at excitation and emission wavelengths of 485 and 538 nm (Fluoroskan Ascent FL, Thermo Fisher Scientific, USA). FITC-dextran concentrations were measured against a standard curve produced by serial dilution of FITC-dextran in mice serum.

## Periodic acid-Schiff staining and histopathological analysis

The mouse colons were removed and fixed in Carnoy's solution for 24 h and subsequently embedded in paraffin. The colon sections were stained with the periodic acid-Schiff staining method. Hepatic histological scoring was conducted by an experienced pathologist at the College of Veterinary Medicine Animal Disease Diagnostic Center (National Chung Hsing University, Taichung, Taiwan).

## Visualized the bacteria co-localized the mucus layer

(i) Paraffin embedding with Carnoy's fixation

The mouse colonic tissues with fecal matter were immersed in Carnoy's solution (60% methanol, 30% chloroform, 10% glacial acetic acid) for 24 h. For embedding, samples were immersed twice in anhydrous methanol for 30 min each, 100% ethanol for 20 min each, and xylene for 15 min each, and finally immersed in paraffin wax (Leica, Germany) at 56–58 °C for 3 h each. Blocks were hardened at room temperature, sectioned to a 4 µm thickness, floated on a water bath at 40–45 °C, and transferred to slides.

(ii) Fluorescent in situ hybridization

Slides were deparaffinized by immersing twice in xylene for 15 min each and 100% ethanol for 5 min each, following which the samples were treated with protease K (5 µg/mL) at 37 °C for 15 min and the treated samples were immersed in 0.9 M NaCl, 20 mM Tris pH 7.4 for 10 min in preparation for fluorescence in situ hybridization (FISH). FISH targeting bacterial cells was performed using the oligonucleotide probe EUB338 (5′-GCTGCCTCCCGTAGGAGT-3′, with a 5′ Alexa 647 label) custom synthesized (Li-Tzung Inc., Taiwan). Sections were incubated in a hybridization buffer (0.9 M NaCl, 0.02 M Tris-HCl pH 7.4, 0.01% SDS, 30% formamide) and 2 µM probe at 46 °C for 2.5 h. Samples were washed at 48 °C for 15 min excess wash buffer after hybridization. (0.215 M NaCl, 0.02 M Tris-HCl pH 7.5, 5 mM EDTA).

(iii) Fluorescent staining with Hoechst 33258 and wheat germ agglutinin (WGA)

Sections were dyed with Hoechst 33258 (10 µg/mL) and WGA (40 µg/mL) Alexa Fluor 488 conjugate (Thermo Fisher Inc., USA) in phosphate-buffered saline for 15 min and incubated twice for 3 min each in wash buffer (0.112 M NaCl, 20 mM Tris-HCl pH 7.4, 5 mM EDTA, 0.01% SDS). Slides were then washed in water and dried or were submerged for 3 min each into 50%, 80%, and 96% (v/v) ethanol and then dried. Slides were subsequently mounted with Prolong anti-fade mounting media (Life Technologies) and placed in a dark environment at room temperature until the mounting medium solidified. Images were acquired with a Leica TCS SP5 II confocal microscope (Joint Center for Instruments and Researches, College of Bioresources and Agriculture, National Taiwan University, Taiwan).

(iv) Measurement of mucus thickness

The mucus layer was determined by using Fiji to measure the width of the region brightly stained with WGA and on the interior of the host epithelium[61]. We visualized the border of the host epithelium using differential interference contrast and then merged the channel to measure the distance.

## Measurement of serum LPS and insulin concentration

Pierce LAL Chromogenic Endotoxin Kit (#88282, Thermo Scientific, USA) was used to detect the LPS levels in the serum. Serum insulin levels were measured using commercial kits supplied by Mercodia AB (#10-1247-01, Mouse Insulin ELISA kit, Mercodia, Sweden).

## Gut microbiota analyses

For these analyses, we decided to use cecal samples instead of fecal samples due to the cecum's pivotal role in microbial fermentation, notable impact on gut microbiota composition, and potential association with metabolic diseases. DNA from cecal samples was extracted according to the manufacturer's instructions using the QIAamp PowerFecal DNA Kit (Qiagen, Netherlands). PCR amplification of 16S rRNA V4 regions was performed using the 515F/806R bacterial primer pair (515F: 5′-GTGCCA GCMGCCGCGGTAA-3′ and 806R: 5′-GGACTACHVGGGTWTCT AAT-3′), with a 50 μL reaction volume containing 25 μL 2X Taq Master Mix (Thermo Scientific, USA), 0.2 M of each forward and reverse primer, and 20 ng DNA template. Thermal cycling consisted of initial denaturation at 95 °C for 5 min, followed by 25 cycles of denaturation at 98 °C for 20 s, annealing at 57 °C for 15 s, and elongation at 72 °C for 30 s, with a final hold at 72 °C for 10 min. Next, amplified products were subjected to 2% agarose gel electrophoresis. The amplified PCR product was attached with Illumina sequencing adapters with a Nextera XT Index kit and then purified using AMPure XP beads. Library quantification was achieved employing the DNA 1000 kit and 2100 Bioanalyzer instrument (Agilent Technologies, USA), and sequencing (single-end reads) was completed using Illumina NextSeq (Illumina, USA). Raw sequences were processed following the QIIME2 pipeline. Raw sequences were quality filtered, trimmed, de-noised, merged, and chimeric sequences removed using the QIIME2 dada2 plugin. The ASVs were aligned against the SILVA database (version 138). The alpha diversity, including the observed ASVs and Shannon and Simpson indices, was computed using the vegan package in R. A PCoA was conducted using the Bray–Curtis distance. Additionally, an ANOSIM was performed to determine the heterogeneity of gut microbiota among the groups. Finally, a heatmap at the genus level was generated using the heatmap3 package in R.

## Statistics and reproducibility

All statistical analyses were carried out using R Studio (version 1.2.5001), R (version 3.6.1), or GraphPad Prism (version 9.5.1). Data are presented as the mean ± standard deviation (s.d.). For experiments with lecithin, sucrose fatty acid esters, and CMC emulsifier-fed mice, statistical analysis was performed using one-way analysis of variance (ANOVA) with the Tukey's range test to determine significant differences between groups, while for experiments with MDG emulsifier-fed mice, a two-tailed Student's $t$-test was used for the same purpose. Heterogeneity of the cecal microbiota among groups in the PCoA was assessed using ANOSIM. The gut microbiome-associated vector in the PCoA was calculated using the "envfit" function in the vegan package in R. Heatmap analysis of the relative abundances of cecal microbiota was conducted using the Kruskal–Wallis test and Wilcoxon signed-rank test for experiments with hydrophilic and lipophilic emulsifier-fed mice, respectively. Spearman's correlation coefficient analysis was conducted to assess the relationship between gut microbiome genera and obesogenic and metabolic biomarkers.

## Reporting summary

Further information on research design is available in the Nature Portfolio Reporting Summary linked to this article.

## Data availability

The raw 16S rRNA sequencing data that support the findings of this study have been deposited in the NCBI Short Read Archive (accession numbers: BioProject, PRJNA944655; BioSample, SAMN33757824). The source data behind the graphs in the paper can be found in Supplementary Data 1. The gut microbiome amplicon sequence variant (ASV) data and taxonomy information can be found in Supplementary Data 2.

## Code availability

The "Methods" section provides a description of the bioinformatics tools, software versions, parameters, and open-source codes utilized in this study. More details regarding codes to reproduce the analyses are available upon request.

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

## Acknowledgements

This study was supported by the Ministry of Science and Technology, Taiwan (109-2327-B-002-005, 109-2314-B-002-103-MY3, 109-2314-B-002-064-MY3, 110-2327-B-002-007, 111-2628-B-002-047, 111-2327-B-002-008, 106-3114-B-002-003, 107-2321-B-002-039, 108-2321-B-002-051, and 107-2321-B-002-017) and the National Science and Technology Council, Taiwan (NSTC 112-2327-B-002-009, 113-2321-B-002-026, and 113-2321-B-002-022). The authors acknowledge the research collaboration

and technical support provided by the National Human Microbiome Core Facility, Taiwan (NSTC 112-2740-B-A49-002).

## Author contributions

S.P. provided instructions and assisted in the experiments, performed bioinformatics and statistical analyses, interpreted the results, and drafted the manuscript; W.-K.W. designed and provided instructions for the experiments, and reviewed and revised the manuscript; C.-T.C. performed the animal experiments; N.W. and H.-C.H. provided technical support in the visualization of bacteria/mucus layer co-localization; Y.-L.L. and S.-P.T. provided instructions and performed the in vitro cellular insulin-resistance experiment; R.-A.C., H.-S.H., and Y.-H.C. assisted in all experiments; P.-Y.L. provided support in the bioinformatics analysis; H.-L.C. provided technical support in mouse experiments; T.-C.D.S., S.-L.T, and C.-T.H. critically reviewed the manuscript; M.-S.W. and L.-Y.S. designed the experiments, acquired funding, and revised the manuscript. All authors had full access to all data and accepted responsibility to submit the manuscript for publication.

## Competing interests

The authors declare no competing interests.
