## [Peer review file · Communications Biology]

Reviewers' comments:

Reviewer #1 (Remarks to the Author):

Purpose of the study is to evaluate the safety of a variety of common food additives. This study examines the impact of several emulsifiers on the gut lining, gut microbiota, and on a variety of metabolic endpoints. The translational relevance is high and of potentially high impact. The manuscript is well-written and the findings are novel. The authors present a good coverage of the challenges raised by the lack of a universal classification of dietary additives such as emulsifiers. They also contextualize with some epidemiological information about exposure is helpful in this emerging area. Overall, there is a lot of enthusiasm for this work and for the point that is being made very clearly here. There are some issues with the paper that somewhat reduce its impact that could be relatively easily addressed.

Points to address:

- The authors appear to have divided the emulsifiers into categories even at the level of the Introduction. Was this done before the study or were they grouped post hoc after it was clear that the different emulsifiers had different effects on the dependent measures? If it was the former, then it would help to justify up front why you expected the additives in the different categories to have different effect profiles. If it was the latter, then I would rather the introduction not divide them. This would be more a topic for the Discussion once it was clear that these unique effects by category were obtained.
- Again, I really like the point that there is no universal classification of the emulsifiers (lines 79-85). I am not sure that the complicated panels shown in Fig 1 adds anything else to the text. I would remove this distracting figure.
- Mice were purchased from a national source in Taiwan, but they are said to be C57BL/6J mice. Does the "J" not mean that the mice were obtained from Jackson Labs? Are these Jackson-derived mice that are bred in Taiwan for fewer than 20 generations? If not, then they are a separate sub-strain. Please clarify.
- How are the mg/kg dosages of the emulsifiers given in food calculated given that these animals are group-housed? Please clarify dosages. Relatedly, line 247 indicates that mice were limited to 85kcal/week – how was this done?
- Some of the emulsifiers were given in water and some in food. Would it not be expected that some of the apparent differences between them might result from the different routes of administration? It is also the case that the hydrophilic emulsifiers were given for 17 wks with a 12 h fast at the end, while the lipophilic emulsifiers were fed for 14 wks and no fasting is mentioned. The doses of at least some of the lipophilic substances, like MDG, are much higher than the concentrations given in water. All of these could impact the results and diminish what can be concluded about the comparisons among the overall "classes" of emulsifiers. These issues do not undercut the most important point, however, which is that the safety of these emulsifiers need to be reassessed.
- Authors should justify why they used only cecal samples for the 16S analysis. There is so much variation in the literature about how the gut microbial samples are collected, so it would be good to be very clear about how it was done and why as well as perhaps the pros and cons of the method used.
- The statistics section in the Methods isn't very helpful. It only mentions a small subset of the statistical tools used and reported.
- The final Results section on MDG should be revised. There was no significant decrease in bacterial distance to epithelial cells and nothing that should even be called a trend. Most of these data are not

significant and should not be presented as though they are (or are close).

- The Discussion should downplay the comparison among the emulsifiers, which may or may not be valid given the points made above. This undermines the real impact of the study, which was not to compare the emulsifiers to one another. It is fine to discuss that they may have very different impacts – that is clear. What is also clear is that most of them should probably not be GRAS! The overall Discussion length can be reduced if these comparisons among treatments are reduced.
- The Discussion is too long with too much coverage of individual microbial changes resulting from each emulsifier. Can this be simplified and reduced some?

Minor points:

- Scientific data never “prove” anything – they support or refute hypotheses. Please revise line 33 in the abstract accordingly.
- Intro line 57: Please add the word “in” before rodents to make this sentence make sense. Are these healthy rodents or a line used to model colitis? It would help to specify.
- Intro line 109: LPS is translocated into the circulation or the circulatory system, not the circulation system. Please correct.
- Fig. 3h – the blue and red signals are difficult to see.
- Line 186 in Results – I assume that there should be a “respectively” added to the end of the sentence?
- The abbreviations are overused in the Results. It is difficult to keep them straight even for readers that are somewhat familiar with the area; much more so if the reader is a newcomer. This undermines the impact of the work. Spell out more of the abbreviations or re-define them in major headings or subsections.
- Line 349-350 – this group has also published that CMC and P80 alter gene expression in mouse brain. This reference should be added. Sci Rep 2022 Jun 1;12(1):9146.
doi: 10.1038/s41598-022-13021-7. Dietary emulsifier consumption alters gene expression in the amygdala and paraventricular nucleus of the hypothalamus in mice Amanda R Arnold 1, Benoit Chassaing 2, Bradley D Pearce 3, Kim L Huhman 4 PMID: 35650224 PMCID: PMC9159048 DOI: 10.1038/s41598-022-13021-7

Reviewer #2 (Remarks to the Author):

In this manuscript, the authors summarized and discussed the safety of dietary emulsifiers by examining their impact on gut microbiota and their relation with obesity and metabolic diseases. On this basis, they investigated the impact of hydrophilic (LEC, SUC, and CMC) and lipophilic (MDG) emulsifiers on the development of obesity and metabolic disease through gut microbiota and host interaction, including gut microbiota dysbiosis, changes in the mucus layer, intestinal permeability, and the translocation of gut-derived LPS into the circulation system. But in this article, some problems remain.

Major comments:

1. It is uncertain about the novelty of the research questions as the effect on gut microbiota is an established part of dietary emulsifiers.
2. Seven emulsifiers were analyzed in Fig. 1, and three of them were selected in Fig. 2. Then what is the basis for the selection?
3. Only one lipophilic emulsifier, MDG, was studied in this study, and its typicality was not clear, which may affect the reliability of the relevant conclusion. It would be better to include more lipophilic

emulsifiers.

4. I noticed that the author mentioned "all dietary emulsifiers were supplemented in drinking water. Mice in each experimental group were allowed free access to food and water for 17 week" in the first part of METHODS. It is not clear to me how to ensure that each mouse received a corresponding dose of emulsifier every day.

5. It is generally assumed that the 8-week-old mice are considered to be adult mice. What is the basis for the choice of 15-week-old mice? And I wonder if the mice were too old after 17 weeks of experiment.

6. The number of animals used in the experiment is confusing. In Fig 2 and Fig 3, the number in each group was different. In Fig 3a-c, 3f, 3g, 3i and 3j, the number of animals in each group was different, neither. In Fig 4c, 4f, 4g, and 4i-k, the number of mice in MDG group and CON group was different. And the same problem goes for Fig 5a-c, 5f, 5g, 5i and 5j. Please explain in detail how to determine the number of mice in the experiment and why the number of mice changed between groups.

7. In the calculation of the dose for mice, it is not clear to me how the authors determined "The dose of this experiment is mimicked as ten times daily exposure in humans" in line 485 on page 21. According to the authors citing reference 10, "CMC has not been extensively studied but is deemed 'generally regarded as safe (GRAS)' and used in various foods at up to 2.0%^{3,6}." Benoit Chassaing selected CMC with a safe dose for human exposure. Please add a full explanation for the dosage of three emulsifiers.

8. The authors found that the mice gained weight and dysregulation of insulin homeostasis after the intervention at week 17. Research has shown that obesity caused by high fat diet can cause insulin homeostasis (Kumar A, et al. High-fat diet-induced upregulation of exosomal phosphatidylcholine contributes to insulin resistance. *Nat Commun.* 2021 Jan 11; 12(1):213. doi: 10.1038/s41467-020-20500-w.). So is insulin homeostasis a direct result of the emulsifier or is the mice gaining weight?

9. Line 240, MDG as an intervention. Please explain the basis for the selection of MDG and its dosage. Is 5.5% the normal dose of MDG or 10 times the dose of human exposure? Why did the intervention last only 14 weeks, which was significantly lower than 17 weeks of intervention in another part of the experiment? In line 248, the author mentioned the intervention lasting 17 weeks, but describe the body weight change at 14 weeks, and it is really unclear to me.

Minor comment:

1. The sentence "A dysfunctional mucus layer has been found in patients with colitis and rodents" in line 56 on page 3 is not accurate enough and may cause misunderstanding. It is recommended to change it to "A dysfunctional mucus layer has been found in colitis in rodents and humans."

2. In line 160 of the article, "whereas the CON, LEC, and CMC were relatively similar to the control." I guess the author made a mistake here.

3. In the description of Fig.2c on Page 6 in result part, there is no significant difference between SUC and CON weight gain, and thus the sentence " At week 17th, SUC and CMC substantially increased the weight gain compared to the CON group (P=0.1563 and P=0.0322, respectively) "in line 124 on page 6 seemed to be inaccurate summary. And the same thing goes for Fig.5b on page 11 and Fig.5g on page 13.

4. In Fig. 3b, there is no change in the Shannon diversity index of the SUC. However, the author mentioned "substantially lowered the Shannon diversity index " in line 149.

Reviewer #3 (Remarks to the Author):

The manuscript by Panyod and colleagues examined the safety of dietary emulsifiers, predominantly through the role of gut microbiota. In this study, the authors showed that hydrophilic emulsifiers such as SUC and CMC induced hyperglycemia and hyperinsulinemia, whereas lipophilic emulsifier MDG impaired circulating lipid and glucose metabolism. The authors further demonstrated that both hydrophilic and lipophilic induced gut microbiota dysbiosis. The authors also showed that hydrophilic emulsifiers have no impact on mucus–bacterial interactions, whereas MDG tended to cause bacterial encroachment into the inner mucus layer and enhance inflammation potential by raising circulating lipopolysaccharide.

While the purpose of the investigation could be an important undertaking, unfortunately, critical issues are pointed out throughout the manuscript. Most importantly, the biological conclusions of this study lack originality. Chassaing B et al. (Nature 2015) have shown the impact of dietary emulsifiers on chronic intestinal inflammation as well as metabolic syndrome, and the mechanisms (dysbiosis of microbiota, microbial encroachment, mucus changes, dysregulated intestinal permeability) by using germ-free mice, TLR5ko and IL-10ko mice. Subsequently, the same and other groups published serial publications demonstrating the detrimental effect of dietary emulsifiers and their impact on microbiota in many ways (cf. Siena MD et al. Nutrients 2022; Viennois E et al. Cell Reports 2020; K Khoshbin K et al. AJP 2020).

In this study, the authors tried to classify the dietary emulsifiers into 2 categories: hydrophilic and hydrophobic (lipophilic). As generally known, the results in this study indicated dietary emulsifiers induced microbial dysbiosis and metabolic abnormality. Since mono and di-glycerides are lipid metabolites, some differences in metabolic indicators were observed between hydrophilic and hydrophobic emulsifier-treated mice. However, no conceptual advances were determined in the context of emulsifier-induced metabolic abnormality (As summarized in Fig. 6). Additionally, the authors pointed out the issue regarding the international classifications of dietary emulsifiers in Fig. 1. Indeed, this rationale could be very important in public health and food safety since the food system has become global in the last few decades. Nonetheless, this is out of the scope of the present journal in biology and should be precisely discussed in specific journals in the field of public health and food safety. The author's experimental approaches employing mice studies and several emulsifiers actually do not have a direct connection with this rationale.

Specific comments:

Line1: The title 'through the lens of gut microbiota' is not mechanistic and is like a title for review articles. This should be corrected.

Line103-105: As above the effect of emulsifiers on metabolic syndrome has been already reported. The rationale for reassessing the safety of emulsifiers is not clear. The authors should show the mechanism how the differences between hydrophilic and hydrophobic emulsifiers affect the metabolic outcome.

Line486: The authors used ten times daily exposure in humans for LEC and SUC. Is this based on the average exposure? Is there any previous evidence?

Line501: In the MDG diet, soybean oil was replaced with MDG. The authors should add the table demonstrating nutrition facts in the MDG diet together with AIN-93M diet.

Figure 1: Although the contents of this figure could be interesting, the data seems not to be original. Instead, the authors summarized the classifications of emulsifiers in each organization, and they nicely re-constructed the data for ref#13 and #9. It could be a main figure in public health-type review articles. However, it should go to the supplementary table in the original article in biology journals.

Point-by-point response to reviewers' comments

Reviewer #1 (Remarks to the Author):

Purpose of the study is to evaluate the safety of a variety of common food additives. This study examines the impact of several emulsifiers on the gut lining, gut microbiota, and on a variety of metabolic endpoints. The translational relevance is high and of potentially high impact. The manuscript is well-written and the findings are novel. The authors present a good coverage of the challenges raised by the lack of a universal classification of dietary additives such as emulsifiers. They also contextualize with some epidemiological information about exposure is helpful in this emerging area. Overall, there is a lot of enthusiasm for this work and for the point that is being made very clearly here. There are some issues with the paper that somewhat reduce its impact that could be relatively easily addressed.

Response:

We are grateful for your valuable feedback on our manuscript. As per your suggestion, we have revised our manuscript to address the highlighted issues and improve its impact.

Points to address:

• The authors appear to have divided the emulsifiers into categories even at the level of the Introduction. Was this done before the study or were they grouped post hoc after it was clear that the different emulsifiers had different effects on the dependent measures? If it was the former, then it would help to justify up front why you expected the additives in the different categories to have different effect profiles. If it was the latter, then I would rather the introduction not divide them. This would be more a topic for the Discussion once it was clear that these unique effects by category were obtained.

Response:

We appreciate your feedback regarding mentioning the division of the studied emulsifiers into categories in the Introduction. To clarify, emulsifiers were categorized into hydrophilic and lipophilic groups after initiating the study because we realized that mono- and diglycerides (MDGs) could not be dissolved in water, unlike the other emulsifiers, which demonstrated hydrophilic properties. For the sake of clarity, we have incorporated this explanation in the revised Introduction.

Lines 102 to 105: "Moreover, at the beginning of the study, it became evident that MDGs could not be dissolved in water, unlike the remaining emulsifiers. Consequently, we categorized the selected emulsifiers as hydrophilic and lipophilic to better accommodate their distinct properties and experimental requirements."

• Again, I really like the point that there is no universal classification of the emulsifiers (lines 79-85).
I am not sure that the complicated panels shown in Fig 1 adds anything else to the text. I would
remove this distracting figure.

**Response:**

Thank you for your comment. We have removed Figure 1 from the main figures and
incorporated it in the revised Supplementary Information as Supplementary Figure 1.

• Mice were purchased from a national source in Taiwan, but they are said to be C57BL/6J mice.
Does the “J” not mean that the mice were obtained from Jackson Labs? Are these Jackson-derived
mice that are bred in Taiwan for fewer than 20 generations? If not, then they are a separate sub-strain.
Please clarify.

**Response:**

We appreciate your inquiry regarding the origin of the C57BL/6 mice used in our study. We
have contacted the National Laboratory Animal Center (Taipei, Taiwan) to clarify this matter. Upon
confirmation, we have verified that these mice are, in fact, C57BL/6JNarl mice, a sub-strain that has
been bred and maintained at the National Laboratory Animal Center in Taiwan. We sincerely
apologize for any confusion that may have arisen from this oversight. We have corrected the name
of the mouse strain in the revised manuscript to accurately reflect its origin.

• How are the mg/kg dosages of the emulsifiers given in food calculated given that these animals are
group-housed? Please clarify dosages. Relatedly, line 247 indicates that mice were limited to
85kcal/week – how was this done?

**Response:**

Thank you for your pertinent questions. For hydrophilic emulsifier-fed mice, emulsifiers were
administered in the drinking water, and their dosages were calculated based on the daily emulsifier
consumption per mouse and the average body weight of mice within each experimental group. To
clarify this, we have incorporated the following sentence in the revised manuscript:

Lines 467 to 468: “Emulsifier dosages were calculated based on the average body weight of
mice within each experimental group.”

Regarding the diet limitation to 85 kcal/week, this was achieved by providing a controlled
amount of food to each mouse to ensure that their weekly caloric intake did not exceed 85 kcal. To
this end, we provided food equivalent to 12.14 kcal/mouse/day (12.14 kcal × 7 days = 85 kcal/week).
To clarify this, we have added this information to the revised manuscript:

Lines 485 to 486: Mice were limited to 85 kcal/week¹⁶, receiving 12.14 kcal/mouse daily.

• Some of the emulsifiers were given in water and some in food. Would it not be expected that some
of the apparent differences between them might result from the different routes of administration? It
is also the case that the hydrophilic emulsifiers were given for 17 wks with a 12 h fast at the end,
while the lipophilic emulsifiers were fed for 14 wks and no fasting is mentioned. The doses of at least
some of the lipophilic substances, like MDG, are much higher than the concentrations given in water.
All of these could impact the results and diminish what can be concluded about the comparisons
among the overall “classes” of emulsifiers. These issues do not undercut the most important point,
however, which is that the safety of these emulsifiers need to be reassessed.

**Response:**

We appreciate your insightful comments regarding the different forms of administration and
exposure durations of emulsifiers. We originally aimed to compare the effects of lecithin, sucrose
fatty acid esters, CMC, and MDGs. However, at the initial stage of the study, it became evident that
MDGs could not be dissolved in water, unlike the other emulsifiers, which forced us to administer
MDGs in the food. We agree that the two forms of administration of emulsifiers (drinking water and
food) can introduce variability and contribute to the observed differences in outcomes. These different
forms of administration can impact factors such as the rate of absorption, metabolism, and
bioavailability of emulsifiers, potentially influencing their effects on the body. Thus, we have
acknowledged this potential source of variability in the revised manuscript.

Lines 426 to 431: “Moreover, we utilized two different forms of emulsifier administration,
namely drinking water and dietary supplementation. This approach may have contributed to the
observed differences in outcomes, making it challenging to directly compare the effects of
hydrophilic and lipophilic emulsifiers. In future studies aiming to compare the effects of these two
types of emulsifiers, both should be administered through dietary supplementation to minimize
potential sources of variability.”

Regarding fasting at the end of the different treatments, both hydrophilic and lipophilic
emulsifier-fed mice underwent a 12 h fasting period before sacrifice. We apologize for neglecting to
mention this for the lipophilic emulsifier-fed group. We have added this information to the section
describing the protocol for lipophilic emulsifier-fed mice in the revised manuscript.

Line 490: “Before sacrifice, mice fasted for 12 h.”

As for your comment on the MDG dose provided in the food, we acknowledge that it was
notably higher than those of hydrophilic emulsifiers given in the drinking water. These differences
were taken into consideration during the experimental design. Our primary objective was to

investigate the effects of these emulsifiers on gut microbiota composition and metabolic biomarkers,
and we recognize that different administration forms and doses may influence outcomes. However,
although these differences may have affected direct comparisons between the two emulsifier
categories, as you mentioned, our study still emphasizes the importance of reassessing the safety of
these emulsifiers, highlighting potential concerns related to their consumption.

• Authors should justify why they used only cecal samples for the 16S analysis. There is so much
variation in the literature about how the gut microbial samples are collected, so it would be good to
be very clear about how it was done and why as well as perhaps the pros and cons of the method used.

**Response:**

Thank you for your valuable suggestions. We acknowledge the importance of sample
collection methods, particularly for the purpose of 16S analysis in this context. In this study, cecal
contents were collected after sacrificing mice, stored in sterile tubes, and frozen at -80 °C before use.
The decision to use only cecal samples was based on the cecum's pivotal role in microbial
fermentation, notable impact on gut microbiota composition, and potential association with metabolic
diseases. However, we are mindful of the existence of various methods for gut microbial sampling,
each with distinct advantages and limitations. Human cecal microbiota differs both quantitatively and
qualitatively from the fecal counterpart, with significantly fewer anaerobes and more facultative
anaerobes². A study revealed that fecal samples offer significant information on the cecal microbial
diversity, despite the OTUs not accurately reflecting the proportional composition. While
qualitatively akin, cecal and fecal microbiota diverge quantitatively. In addition, fecal samples have
proven to be valuable for detecting shifts in cecal microbiota³. In summary, fecal sampling offers a
non-invasive method and provides information on the overall gut health; however, it may display
temporal variation due to transit time and diet effects. Conversely, cecal sampling provides consistent
information on microbial communities, and insights into microbial diversity and functions;
nevertheless, it is an invasive sampling method, not being suitable for certain study designs. We have
explained our decision to use cecal samples in the revised manuscript.

Lines 569 to 571: "For these analyses, we decided to use cecal samples instead of fecal
samples due to the cecum's pivotal role in microbial fermentation, notable impact on gut microbiota
composition, and potential association with metabolic diseases."

• The statistics section in the Methods isn't very helpful. It only mentions a small subset of the
statistical tools used and reported.

**Response:**

Thank you for your comment. We have revised the statistical analysis section to provide
detailed information on this matter, as follows:

Lines 593 to 605: **“Statistics and reproducibility**

All statistical analyses were carried out using R Studio (version 1.2.5001), R (version 3.6.1),
or GraphPad Prism (version 9.5.1). Data are presented as the mean \pm standard deviation (s.d.). For
experiments with hydrophilic emulsifier-fed mice, statistical analysis was performed using one-way
analysis of variance (ANOVA) with the Tukey's range test to determine significant differences
between groups, while for experiments with lipophilic emulsifier-fed mice, a two-tailed Student's *t*-
test was used for the same purpose. Heterogeneity of the cecal microbiota among groups in the PCoA
was assessed using ANOSIM. The gut microbiome-associated vector in the PCoA was calculated
using the “envfit” function in the vegan package in R. Heatmap analysis of the relative abundances
of cecal microbiota was conducted using the Kruskal–Wallis test and Wilcoxon signed-rank test for
experiments with hydrophilic and lipophilic emulsifier-fed mice, respectively. Spearman's
correlation coefficient analysis was conducted to assess the relationship between gut microbiome
genera and obesogenic and metabolic biomarkers.”

• The final Results section on MDG should be revised. There was no significant decrease in bacterial
distance to epithelial cells and nothing that should even be called a trend. Most of these data are not
significant and should not be presented as though they are (or are close).

**Response:**

We appreciate your comment on the data interpretation. Accordingly, we have revised the
interpretation of the mentioned results to highlight the lack of significance or trend.

Line 315 to 327: **“MDG slightly decreased the distance of bacteria to epithelial cells and**
**enhanced inflammation potential by increasing circulating LPS**

Next, we examined how lipophilic emulsifier MDG affected the physiological change in the
colon, intestinal permeability, and translocation of the bacterial product. The colon length of mice in
the MDG-fed group was not significantly different from that in the control group (Fig. 4f). The PAS-
stained colon sections and colonic epithelial damage histological score demonstrated that MDG did
not induce colitis (Supplementary Fig. 6). Furthermore, although the distance of bacteria to IECs was
shorter in mice fed the MDG diet than in those fed the control diet, this difference was not statistically
significant ($P = 0.1780$) (Fig. 4g, h), suggesting that the MDG-altered gut microbiome may enter the
inner layer of the mucus. The reduced distance between bacteria and epithelial cells may promote gut

inflammation-associated disease. Additionally, MDG did not affect FITC-dextran-based intestinal
permeability (Fig. 4i). However, the serum LPS level was significantly elevated in mice fed MDG
(Fig. 4j), suggesting that the MDG-altered microbiota increased LPS production.”

• The Discussion should downplay the comparison among the emulsifiers, which may or may not be
valid given the points made above. This undermines the real impact of the study, which was not to
compare the emulsifiers to one another. It is fine to discuss that they may have very different impacts
– that is clear. What is also clear is that most of them should probably not be GRAS! The overall
Discussion length can be reduced if these comparisons among treatments are reduced.

**Response:**

Thank you for your insightful comment. We initially aimed to compare the effects of various
emulsifiers; however, the differences in administration, doses, and other variables hindered these
comparisons. Thus, we agree that it is essential to emphasize that the main objective of our study was
to investigate the individual and collective impacts of these emulsifiers on gut microbiota
composition and metabolic biomarkers rather than make head-to-head comparisons. Accordingly, we
have reduced the length of the Discussion section by minimizing inter-emulsifier comparisons and
changed its focus to the broader implications of our findings. We fully agree that the core message of
our study is the urgent need for a reassessment of the safety of these emulsifiers, given their potential
to disrupt gut microbiota and metabolic health. Thus, the revised Discussion section has been
streamlined, enhancing the clarity and impact of our key findings.

• The Discussion is too long with too much coverage of individual microbial changes resulting from
each emulsifier. Can this be simplified and reduced some?

**Response:**

Thank you for your pertinent comment. As suggested, the length of the Discussion section
has been reduced by summarizing information on the association of gut microbiota taxa found
differently abundant in our study and health/disease-related factors in Supplementary Table 1 (see
below). This change simplified and reduced the length of the revised Discussion, while still providing
comprehensive information on microbial alterations.

**Supplementary Table 1. Comprehensive information on the association of gut microbiota taxa**
 **whose abundance changed in groups treated with hydrophilic/lipophilic emulsifiers with**
 **health- and disease-related factors.**

Taxon	Health/disease-related factors	References
Akkermansia muciniphila	Pasteurized A. muciniphila supplementation enhances insulin sensitivity and decreases insulinemia, plasma total cholesterol levels, and obesity in overweight/obese insulin-resistant volunteers.	3
Atopostipes	Atopostipes abundance is enriched in mice with carbon tetrachloride-induced hepatic injury.	4
Blautia	Blautia abundance is enriched in patients with NASH, and is associated with increased lipopolysaccharide levels, visceral fat accumulation and obesity in adults, and increased blood insulin levels in children.	5-7
Collinsella	Low dietary fiber consumption increases Collinsella abundance, and Collinsella abundance correlates with circulating insulin levels in overweight/obese pregnant women.	8
Coriobacteriaceae UCG-002	Coriobacteriaceae UCG-002 exhibits anti-inflammatory function, and augmented Coriobacteriaceae UCG-002 abundance increases the beneficial bacterial metabolite short-chain fatty acid levels.	9
Clostridium sensu stricto 1	Clostridium sensu stricto 1 abundance is increased in patients with duodenal strictures.	10
Desulfovibrio	Desulfovibrio plays a vital role in NAFLD pathogenesis by increasing intestinal permeability and hepatic CD36 expression.	11
Dubosiella	Dubosiella relative abundance is decreased in mice with DSS-induced colitis, having potential as ulcerative colitis amelioration bacteria.	12
Faecalibaculum rodentium	F. rodentium stimulates epithelial proliferation and turnover by dampening retinoic acid generation that helps intestinal eosinophil survival.	13
Enterobacter aerogenes and Enterobacter cloacae	Enterobacteriaceae is associated with inflammatory bowel disease pathogenesis and progression. E. aerogenes and E. cloacae are found in several outbreaks of hospital-acquired infections.	14,15
Enterorhabdus	Enterorhabdus relative abundance is increased in patients with prediabetes.	16
Eubacterium coprostanoligenes	Eubacterium coprostanoligenes group is largely found in patients with homocystinuria.	17

[Eubacterium] xylanophilum group	[Eubacterium] xylanophilum group abundance is enriched in mice with high salt-induced hypertension.	18
Jeotgalicoccus	Jeotgalicoccus is positively correlated with insulin concentration in diabetic rats.	19
Lachnoclostridium	Lachnoclostridium is associated with obesity, and highly abundant Lachnoclostridium is linked to decreased circulating acetate levels, which are associated with increased visceral fat in a large population-based cohort.	20,21
Lachnospiraceae NK4A136 group	Lachnospiraceae NK4A136 group is a potential probiotic whose abundance is reduced in high-fat diet-fed mice.	22
Lachnospiraceae UCG-006	Lachnospiraceae UCG-006 abundance is increased in high-fat diet-fed mice.	23
Muribaculaceae	Muribaculaceae abundance is enriched in lean mice and potentially involved in complex carbohydrate degradation.	24,25
Muribaculum	Muribaculum potentially maintains mouse gut homeostasis.	26
Olsenella	Olsenella abundance is enriched in lean people.	27
Oscillibacter	Oscillibacter abundance is enriched in diabetic mice fed a high-fat carbohydrate-free diet.	28
Parabacteroides distasonis	P. distasonis alleviates obesity and metabolic dysfunction by producing succinate and secondary bile acid.	29
Parasutterella	Parasutterella potentially plays a role in bile acid maintenance and cholesterol metabolism, and is associated with improved low-density lipoprotein levels in healthy individuals.	30,31
Staphylococcus aureus	S. aureus easily colonizes the infant's intestine due to a poorly competitive gut microbiota community.	32
Streptococcus pyogenes	S. pyogenes causes both non-invasive and invasive illnesses, including nonsuppurative sequelae.	33
Streptococcus gallolyticus	S. gallolyticus colonization is associated with colorectal cancer occurrence.	34
Turicibacter	Turicibacter is more abundant in lean than in obese rodents, and is a potential anti-inflammatory taxon.	35

Minor points:

• Scientific data never “prove” anything – they support or refute hypotheses. Please revise line 33 in
the abstract accordingly.

**Response:**

Thank you for highlighting this. We have replaced the word “proved” with “demonstrates” in
the revised manuscript at the mentioned instance.

Lines 27 to 28: “This study demonstrates that sucrose fatty acid esters and
carboxymethylcellulose induced hyperglycemia and hyperinsulinemia.”

• Intro line 57: Please add the word “in” before rodents to make this sentence make sense. Are these
healthy rodents or a line used to model colitis? It would help to specify.

**Response:**

Thank you for your comment. This has been corrected in the revised manuscript by rephrasing
the sentence as follows:

Lines 47 to 48: “A dysfunctional mucus layer has been found in both murine and human
colitis³⁷.”

• Intro line 109: LPS is translocated into the circulation or the circulatory system, not the circulation
system. Please correct.

**Response:**

Thank you for highlighting this oversight. The sentence has been modified in the revised
manuscript as follows:

Lines 98 to 101: “This study investigated the impact of lecithin, sucrose fatty acid esters,
CMC, and MDGs on the development of obesity and metabolic disease through gut microbiota and
host interactions, including gut microbiota dysbiosis, changes in the mucus layer, intestinal
permeability, and the translocation of gut-derived LPS into the circulatory system.”

• Fig. 3h – the blue and red signals are difficult to see.

**Response:**

Thank you for pointing this out. We have addressed this concern by adjusting the images’
exposure and saturation. Please find below the original Figure 3h and revised Figure 2h (former
Figure 3h) for comparison:

Original Figure 3h:

Revised Figure 2h:

• Line 186 in Results – I assume that there should be a “respectively” added to the end of the sentence?

**Response:**

Thank you for highlighting this oversight. “Respectively” has been added to the end of the
sentence, which has been modified in the revised manuscript as follows:

Lines 186 to 188: “Compared with the control group, the lecithin, sucrose fatty acid esters,
and CMC groups had 18 (9 \uparrow /9 \downarrow), 19 (7 \uparrow /12 \downarrow), and 19 (9 \uparrow /10 \downarrow) significantly different genera (P<0.05),
respectively.”

• The abbreviations are overused in the Results. It is difficult to keep them straight even for readers
that are somewhat familiar with the area; much more so if the reader is a newcomer. This undermines
the impact of the work. Spell out more of the abbreviations or re-define them in major headings or
subsections.

**Response:**

Thank you for your suggestion. To enhance readability and clarity, we have spelled out
several abbreviations and re-defined them in major headings or subsections.

• Line 349-350 – this group has also published that CMC and P80 alter gene expression in mouse
brain. This reference should be added. Sci Rep 2022 Jun 1;12(1):9146.
doi: 10.1038/s41598-022-13021-7. Dietary emulsifier consumption alters gene expression in the
amygdala and paraventricular nucleus of the hypothalamus in mice Amanda R Arnold 1, Benoit
Chassaing 2, Bradley D Pearce 3, Kim L Huhman 4 PMID: 35650224 PMCID: PMC9159048 DOI:
10.1038/s41598-022-13021-7

**Response:**

Thank you for your valuable suggestion. As recommended, we have added this information
to the revised manuscript, citing this reference, and included it in the References section.

Lines 356 to 357: “Moreover, CMC and P80 alter gene expression in mouse brains, which
potentially impacts stress responses²¹.”

Lines 685 to 687: 21 Arnold, A. R., Chassaing, B., Pearce, B. D. & Huhman, K. L. Dietary
emulsifier consumption alters gene expression in the amygdala and paraventricular nucleus of the
hypothalamus in mice. *Sci. Rep.* **12**, 9146 (2022).

**Reviewer #2 (Remarks to the Author):**

In this manuscript, the authors summarized and discussed the safety of dietary emulsifiers by
examining their impact on gut microbiota and their relation with obesity and metabolic diseases. On
this basis, they investigated the impact of hydrophilic (LEC, SUC, and CMC) and lipophilic (MDG)
emulsifiers on the development of obesity and metabolic disease through gut microbiota and host
interaction, including gut microbiota dysbiosis, changes in the mucus layer, intestinal permeability,
and the translocation of gut-derived LPS into the circulation system. But in this article, some
problems remain.

**Response:**

We are grateful for your valuable feedback on our manuscript. As per your suggestion, we
have revised our manuscript to address the highlighted issues and improve its content.

Major comments:

1.It is uncertain about the novelty of the research questions as the effect on gut microbiota is an
established part of dietary emulsifiers.

**Response:**

We greatly appreciate your feedback concerning the novelty of our research questions.
Traditionally, the evaluation and approval of food additives' safety have relied on toxicological
evidence following the Generally Recognized As Safe (GRAS) guidelines. However, recent
discoveries highlighting the significant role of gut microbiota in health and disease have prompted
researchers to shift their focus towards studying the potential adverse health effects of food additives
through the metaorganism–pathogenesis lens and their interactions with the gut microbiota. In light
of this paradigm shift, it has become imperative to explore the safety of food additives within the
context of host–gut microbiota interactions, bridging the existing gaps in the current literature and
providing essential insights to inform food production regulators.

Currently, our understanding of the effects of emulsifiers on the development of metabolic
diseases is limited. While some dietary emulsifiers, such as CMC and P80, have been previously
investigated, this study aimed to reassess the safety of dietary emulsifiers that have received less
attention or are infrequently reported in the literature. These emulsifiers include sucrose fatty acid
esters, lecithin, and MDG.

Our study demonstrated that sucrose fatty acid esters and CMC induced hyperglycemia and
hyperinsulinemia, while MDG disrupted circulating lipid and glucose metabolism. Both hydrophilic
and lipophilic emulsifiers altered the diversity of the intestinal microbiota and induced gut microbiota
dysbiosis. Notably, hydrophilic emulsifiers had no discernible impact on mucus–bacteria interactions.

In contrast, MDG appeared to promote bacterial encroachment into the inner mucus layer and
enhance inflammation, potentially by elevating circulating lipopolysaccharide levels. Thus, our
findings underscore the safety concerns associated with the use of dietary emulsifiers and suggest
that their consumption contributes to the development of metabolic syndromes.

2.Seven emulsifiers were analyzed in Fig. 1, and three of them were selected in Fig. 2. Then what is
the basis for the selection?

**Response:**

Thank you for your pertinent question. A recent report estimated the human dietary exposure
to seven emulsifiers, namely MDGs, lecithin, CMC, sucrose fatty acid esters, P80, stearyl lactylates
(sodium stearyl lactylate and calcium stearyl lactylate), and polyglycerol polyricinoleate, across
two distinct time periods: 1999–2002 and 2003–2010 (Supplementary Figure 1c). Based on the
findings of that study, we opted to focus our investigation on the top four emulsifiers with high
exposure levels, which included MDGs, lecithin, CMC, and sucrose fatty acid esters. The fact that
Figure 2 (now labeled Figure 1 in the revised manuscript) shows the results pertaining to only three
of the four selected emulsifiers is due to our need to divide the emulsifiers into two groups because
of their distinct hydrophilic and lipophilic properties, which required different administration forms.
Consequently, we performed the experiments with each emulsifier group and reported their results
separately. Figs. 1 and 2 show the results of experiments with the three hydrophilic emulsifiers
(lecithin, CMC, and sucrose fatty acid esters), while Figs. 3 and 4 show those with the lipophilic
emulsifier (MDGs). We have clarified the selection of emulsifiers and their categorization into two
groups in the revised manuscript:

Lines 101 to 105: “The selection of these emulsifiers was due to the estimated high dietary
exposure of humans to them¹³. Moreover, at the beginning of the study, it became evident that MDGs
could not be dissolved in water, unlike the remaining emulsifiers. Consequently, we categorized the
selected emulsifiers as hydrophilic and lipophilic to better accommodate their distinct properties and
experimental requirements.”

3.Only one lipophilic emulsifier, MDG, was studied in this study, and its typicality was not clear,
which may affect the reliability of the relevant conclusion. It would be better to include more
lipophilic emulsifiers.

**Response:**

We appreciate your suggestion to include more lipophilic emulsifiers in our study. Due to
limitations of the current study and resource constraints, we were unable to include other lipophilic
emulsifiers in this investigation. However, we acknowledge the importance of focusing on a broader

range of lipophilic emulsifiers in future research to enhance the comprehensiveness of our findings.
This is a valuable input, and we will do our best to address this issue in future studies. We have
mentioned this limitation of our study in the revised manuscript.

Lines 433 to 435: “Additionally, conducting a broader comparison including various types of
hydrophilic and lipophilic emulsifiers could yield more comprehensive and valuable insights.”

4.I noticed that the author mentioned "all dietary emulsifiers were supplemented in drinking water.
Mice in each experimental group were allowed free access to food and water for 17 week" in the first
part of METHODS. It is not clear to me how to ensure that each mouse received a corresponding
dose of emulsifier every day.

**Response:**

Thank you for bringing up this concern. To ensure that mice received the correct dose of
hydrophilic emulsifiers daily, we monitored their liquid intake throughout the experiment. The
lecithin- and CMC-treated groups showed increased liquid intake, while the sucrose fatty acid ester-
treated group did not exhibit a significantly different liquid intake compared to the control group
(Supplementary Figure 2a).

To calculate the actual emulsifier intake per day, we used the theoretical intake. The
theoretical intakes for lecithin and sucrose fatty acid esters were 7523.3 mg/kg bw/day and 1110
360 mg/kg bw/day, respectively. The actual intake for lecithin was calculated as 14931.4 ± 1900.2 mg/kg
bw/day, which was approximately 1.98 times higher than the theoretical intake (Supplementary
Figure 2b). For sucrose fatty acid esters, the actual intake closely resembled the theoretical intake
(Supplementary Figure 2b). As for CMC, which was used as a positive control, we determined its
dosage based on a previous study³⁹ and did not calculate its theoretical intake for this specific
experiment. To clarify these issues, we have included this information in the revised manuscript.

Lines 113 to 122: “All emulsifiers were administered via drinking water. Mice in each
experimental group had unrestricted access to both food and water. The dosage design for lecithin
and sucrose fatty acid esters followed the report on estimated dietary exposure in humans¹³. The
human dosage of emulsifiers was translated to equivalent mouse doses. The doses of lecithin and
sucrose fatty acid esters were 10 times the respective daily exposure levels in humans, that is 7523.3
and 1110 mg/kg bw/day, respectively. CMC (1% in drinking water) dosage was set according to a
previous study and used as a positive control¹⁰. After ingestion of the dietary emulsifier for 17 weeks,
the actual intake for lecithin was nearly double the theoretical intake. In contrast, the actual intake for
sucrose fatty acid esters closely matched the theoretical intake (Supplementary Fig. 2).”

**Supplementary Figure 2. Intake of hydrophilic dietary emulsifiers in mice. a) Liquid intake. b)**

Comparison of theoretical and actual emulsifier intake. Mice were supplemented with or without
 different emulsifiers in drinking water for 17 weeks. Bars express the mean \pm s.d. Statistical analyses
 were performed using one-way ANOVA with the Tukey's range test for comparisons shown as exact
 P-values. CON: control; LEC: lecithin; SUC: sucrose fatty acid esters; CMC:
 carboxymethylcellulose.

5.It is generally assumed that the 8-week-old mice are considered to be adult mice. What is the basis
 for the choice of 15-week-old mice? And I wonder if the mice were too old after 17 weeks of
 experiment.

**Response:**

Thank you for your questions. The selection of 15-week-old mice for this study was based on
 the findings of Chassaing et al., who reported that emulsifier-induced metabolic syndrome is more
 prominently observed in older mice³⁹.

As for the concern about mice being too old at the end of the experiment, we recognize this
 possibility. However, emulsifier treatment needed to last for enough time to allow potential metabolic
 changes to manifest, and the chosen treatment duration (17 weeks) aligns with those in previous
 studies examining similar outcomes.

6.The number of animals used in the experiment is confusing. In Fig 2 and Fig 3, the number in each
 group was different. In Fig 3a-c, 3f, 3g, 3i and 3j, the number of animals in each group was
 different,neither. In Fig 4c, 4f, 4g, and 4i-k, the number of mice in MDG group and CON group was
 different. And the same problem goes for Fig 5a-c, 5f, 5g, 5i and 5j. Please explain in detail how to

determine the number of mice in the experiment and why the number of mice changed between
groups.

**Response:**

Thank you for highlighting this issue. Indeed, data depicted in our figures are from groups
with different numbers of animals. While all general measurements and serum biochemistry analyses
(body weight; relative fat mass; serum total cholesterol, triglyceride, blood glucose, insulin, and LPS
levels; OGTT; HOMA-IR; FITC-dextran concentration; and colon length) were conducted on all
mice in both the hydrophilic and lipophilic experiments (n = 14–15/group), other more specific
analyses, such as the gut microbiota analysis (n = 6–8/group) and immunostaining for bacteria–
intestinal epithelial cell interaction analysis (n = 5–7/group), were done on a randomly selected subset
of mice from each group. The smaller sample size used for some analyses was primarily dictated by
the nature of the sampling process and availability of samples at the time of analysis.

7.In the calculation of the dose for mice, it is not clear to me how the authors determined “The dose
of this experiment is mimicked as ten times daily exposure in humans” in line 485 on page 21.
According to the authors citing reference 10, "CMC has not been extensively studied but is deemed
‘generally regarded as safe (GRAS)’ and used in various foods at up to 2.0%^{3,6}." Benoit Chassaing
selected CMC with a safe dose for human exposure. Please add a full explanation for the dosage of
three emulsifiers.

**Response:**

Thank you for your comment. In this study, we treated mice with daily dosages of emulsifiers
that were approximately 10 times the respective daily exposure levels for humans. Noteworthy, this
translation method involves certain assumptions and limitations. However, this approach aligns with
established practices in the field and serves as an initial step to investigate the effects of these
emulsifiers on mice in a controlled experimental setting.

To determine the dosages of lecithin and sucrose fatty acid esters, we considered their
estimated dietary exposure levels in humans, which were 61 mg/kg bw/day and 9 mg/kg bw/day,
respectively¹. These human exposure levels were then translated to equivalent mouse doses⁴⁰,
calculated as 7523.3 mg/kg bw/day for lecithin and 1110 mg/kg bw/day for sucrose fatty acid esters.
Regarding CMC, we adopted the dosage used in the study by Chassaing et al.³⁹, where CMC was
used as a positive control. We have provided a more detailed explanation of emulsifier dosage
selection in the revised manuscript as follows:

Lines 456 to 474: **“Hydrophilic dietary emulsifiers-fed C57BL/6 mouse model**

The animals were handled following the guidelines of the Institutional Animal Care and Use
Committee of National Taiwan University (approval number: NTU107-EL-00121). Male

C57BL/6JNarl mice were purchased from the National Laboratory Animal Center (Taipei, Taiwan).
The mice were housed in an animal room with a 12 h light/dark cycle at 23 ± 2 °C. After acclimation,
fifteen-week-old mice were randomly divided into four experimental groups (n = 3–4/cage): (i)
control, (ii) lecithin (7523.3 mg/kg bw/day, 4.625% in drinking water), (iii) sucrose fatty acid esters
(1110 mg/kg bw/day, 0.68% in drinking water), and (iv) CMC (1% in drinking water). CMC was
used as a positive control, and its dosage was selected based on a previous study¹⁰. The dosages of
lecithin and sucrose fatty acid esters were selected based on a report on the estimated dietary exposure
of humans (lecithin: 61 mg/kg bw/day; sucrose fatty acid esters: 9 mg/kg bw/day)¹³ and translated to
the corresponding mouse dosages⁵⁹. Emulsifier dosages were calculated based on the average body
weight of mice within each experimental group. Emulsifier doses in this experiment were 10 times
the respective daily exposure levels in humans. Emulsifiers were purchased from Gemfont
Corporation (Taipei, Taiwan). Mice were fed with a normal chow diet (MFG; Oriental Yeast Co.,
Ltd., Tokyo, Japan), and all dietary emulsifiers were supplemented in drinking water. Mice in each
experimental group were allowed free access to food and water for 17 weeks. Before sacrifice, the
mice fasted for 12 h. The mice were subsequently sacrificed using CO₂ asphyxiation. Blood was
collected via cardiac puncture using a syringe. Organs were collected for subsequent analysis.”

8.The authors found that the mice gained weight and dysregulation of insulin homeostasis after the
intervention at week 17. Research has shown that obesity caused by high-fat diet can cause insulin
homeostasis (Kumar A, et al. High-fat diet-induced upregulation of exosomal phosphatidylcholine
contributes to insulin resistance. Nat Commun. 2021 Jan 11; 12(1):213. doi: 10.1038/s41467-020-
20500-w.). So is insulin homeostasis a direct result of the emulsifier or is the mice gaining weight?

**Response:**

Thank you for your pertinent question. While we observed changes in body weight and insulin
homeostasis upon emulsifier administration, we recognize that these outcomes can result from other
factors, including dietary composition. Teasing apart the direct effects of emulsifiers from those of
dietary components is complex, and further research is needed to understand the specific mechanisms
disrupting insulin homeostasis. We have addressed this limitation of our study in the revised
manuscript as well as included the suggested reference.

Lines 424 to 426: “A high-fat diet is widely recognized as a contributor to insulin resistance⁵⁸.
The exact contribution of emulsifiers to insulin homeostasis, and whether the latter is influenced
primarily by emulsifier intake or weight gain warrant further investigation.”

Lines 774 to 775: “58 Kumar, A. *et al.* High-fat diet-induced upregulation of exosomal
phosphatidylcholine contributes to insulin resistance. *Nat. Commun.* **12**, 213 (2021).”

9.Line 240, MDG as an intervention. Please explain the basis for the selection of MDG and it's
dosage. Is 5.5% the normal dose of MDG or 10 times the dose of human exposure? Why did the
intervention last only 14 weeks , which was significantly lower than 17 weeks of intervention in
another part of the experiment? In line 248, the author mentioned the intervention lasting 17 weeks,
but describe the body weight change at 14 weeks, and it is really unclear to me.

**Response:**

Thank you for your questions. The rationale behind including MDGs, a type of fatty acids
composed of mono- and di-acylglycerol, in our study stems from their prevalence in various food and
animal feed products. MDGs are commonly found in dietary sources. In animal feed products,
especially in diets where soybean oil serves as the primary lipid source for energy, MDGs are
frequently included. Given that both soybean oil and MDGs provide energy to the body, we opted to
substitute soybean oil with MDGs in the diet provided to the MDG group to assess their potential
effects.

In terms of dosage selection, we referred to a previous study that utilized 5.5% diacylglycerol
(DAG) as experimental feed. Importantly, DAG is structurally related to MDGs, and this dosage was
considered high in the context of the study. Moreover, in that study, 5.5% DAG was administered for
24 months to a rat model without inducing carcinogenesis⁴². Therefore, we used this MDG dosage in
our study.

Regarding the different durations of lipophilic (14 weeks) and hydrophilic (17 weeks)
emulsifier treatments, this discrepancy is attributed to the distinct experimental protocols designed
for each emulsifier group. In experiments with hydrophilic emulsifiers, the intervention period lasted
for 17 weeks, and all data from these experiments pertain to the 17th week, while in the experiments
with the lipophilic emulsifier, the intervention duration indeed spanned 14 weeks. Noteworthy, our
primary objective was not to do a head-to-head comparison of the effects of hydrophilic and lipophilic
emulsifiers. Rather, we aimed to investigate each emulsifier individually relatively to the control,
using experimental designs and dosage selection approaches tailored to each emulsifier's unique
properties.

We have clarified these issues in the revised manuscript.

Lines 248 to 252: "The dosage of MDG used in this experiment was based on that of
diacylglycerol used in a prior study, showing no signs of systemic toxicity¹⁵. As both soybean oil and
MDG are fats that provide energy to the body, and the control diet contained triacylglycerol mainly
derived from soybean oil, we substituted soybean oil with MDG in the diet fed to the MDG group."

Minor comment:

1.The sentence “A dysfunctional mucus layer has been found in patients with colitis and rodents” in
line 56 on page 3 is not accurate enough and may cause misunderstanding. It is recommended to
change it to“A dysfunctional mucus layer has been found in colitis in rodents and humans.”

**Response:**

Thank you for highlighting this oversight. We have modified this sentence in the revised
manuscript to avoid confusion.

Lines 47 to 48: “A dysfunctional mucus layer has been found in both murine and human
colitis⁷.”

2.In line 160 of the article, "whereas the CON, LEC, and CMC were relatively similar to the control."
I guess the author made a mistake here.

**Response:**

Thank you for bringing this up. We have modified this sentence in the revised manuscript as
follows:

Lines 165 to 167: “The sucrose fatty acid esters group displayed a distinct separation from the
control group counting on the X-axis (PCoA1; 27.32 %), whereas the lecithin and CMC groups were
relatively similar to the control.”

3.In the description of Fig.2c on Page 6 in result part, there is no significant difference between SUC
and CON weight gain, and thus the sentence “ At week 17th, SUC and CMC substantially increased
the weight gain compared to the CON group (P=0.1563 and P=0.0322, respectively) ”in line 124 on
page 6 seemed to be inaccurate summary. And the same thing goes for Fig.5b on page 11 and Fig.5g
on page 13.

**Response:**

Thank you for highlighting these discrepancies. We have modified these sentences in the
revised manuscript to accurately reflect the results.

Lines 123 to 125: “At week 17, the CMC-treated group had increased weight gain compared
to the control group (P = 0.0322), while the group treated with sucrose fatty acid esters showed only
a tendency to gain weight (P = 0.1563) (Fig. 1c).”

Lines 270 to 272: “MDG did not impact the observed ASVs or Shannon diversity index but
reduced the microbiota community evenness and Simpson’s diversity index (P = 0.0285) (Fig. 4a-c).”

Lines 321 to 324: “Furthermore, although the distance of bacteria to IECs was shorter in mice
fed the MDG diet than in those fed the control diet, this difference was not statistically significant (P

= 0.1780) (Fig. 4g, h), suggesting that the MDG-altered gut microbiome may enter the inner layer of
the mucus.”

4.In Fig. 3b, there is no change in the Shannon diversity index of the SUC. However, the author
mentioned “substantially lowered the Shannon diversity index ” in line 149.

**Response:**

Thank you for bringing this up. The mentioned claim has been omitted from the revised
manuscript.

**Reviewer #3 (Remarks to the Author):**

The manuscript by Panyod and colleagues examined the safety of dietary emulsifiers,
predominantly through the role of gut microbiota. In this study, the authors showed that hydrophilic
emulsifiers such as SUC and CMC induced hyperglycemia and hyperinsulinemia, whereas lipophilic
emulsifier MDG impaired circulating lipid and glucose metabolism. The authors further
demonstrated that both hydrophilic and lipophilic induced gut microbiota dysbiosis. The authors also
showed that hydrophilic emulsifiers have no impact on mucus–bacterial interactions, whereas MDG
tended to cause bacterial encroachment into the inner mucus layer and enhance inflammation
potential by raising circulating lipopolysaccharide.

While the purpose of the investigation could be an important undertaking, unfortunately,
critical issues are pointed out throughout the manuscript. Most importantly, the biological conclusions
of this study lack originality. Chassaing B et al. (Nature 2015) have shown the impact of dietary
emulsifiers on chronic intestinal inflammation as well as metabolic syndrome, and the mechanisms
(dysbiosis of microbiota, microbial encroachment, mucus changes, dysregulated intestinal
permeability) by using germ-free mice, TLR5ko and IL-10ko mice. Subsequently, the same and other
groups published serial publications demonstrating the detrimental effect of dietary emulsifiers and
their impact on microbiota in many ways (cf. Siena MD et al. Nutrients 2022; Viennois E et al. Cell
Reports 2020; K Khoshbin K et al. AJP 2020).

In this study, the authors tried to classify the dietary emulsifiers into 2 categories: hydrophilic
and hydrophobic (lipophilic). As generally known, the results in this study indicated dietary
emulsifiers induced microbial dysbiosis and metabolic abnormality. Since mono and di-glycerides
are lipid metabolites, some differences in metabolic indicators were observed between hydrophilic
and hydrophobic emulsifier-treated mice. However, no conceptual advances were determined in the
context of emulsifier-induced metabolic abnormality (As summarized in Fig. 6). Additionally, the
authors pointed out the issue regarding the international classifications of dietary emulsifiers in Fig.
1. Indeed, this rationale could be very important in public health and food safety since the food system
has become global in the last few decades. Nonetheless, this is out of the scope of the present journal
in biology and should be precisely discussed in specific journals in the field of public health and food
safety. The author’s experimental approaches employing mice studies and several emulsifiers actually
do not have a direct connection with this rationale.

**Response:**

We greatly appreciate your comprehensive review of our manuscript and valuable insights.
We have meticulously analyzed your comments and suggestions, and addressed each of them below.

**Lack of originality:** We are familiar with the seminal work by Chassaing et al. and
subsequent studies exploring the impact of dietary emulsifiers on gut microbiota, inflammation, and

metabolic syndrome. We are grateful for the provided references and, after careful scrutiny, we
summarized their research focus:

- • Chassaing et al. investigated the effects of CMC and P80.
- • Siena et al. (*Foods*, 2022, 10.3390/foods11152205) reviewed research on various emulsifiers,
including agar agar, CMC, carrageenan, glycerol monolaurate, gums, lecithins, maltodextrin,
and P80, from in vitro, animal, and human studies.
- • Viennois et al. (*Cell Reports*, 2020, 10.1016/j.celrep.2020.108229) explored the effects of
CMC and P80 on chronic intestinal inflammation.
- • Khoshbin and Camilleri (*Gastrointest Liver Physiol*, 2020, 10.1152/ajpgi.00245.2020)
reviewed the effects of CMC and P80 on intestinal permeability.

Our study aimed to increase this body of knowledge by investigating the effects of specific
hydrophilic (lecithin, CMC, and sucrose fatty acid esters) and lipophilic (MDGs) emulsifiers on the
gut microbiota composition, metabolic biomarkers, and gut barrier integrity. We acknowledge the
existing overlap between the content of the abovementioned studies and that of our study, particularly
regarding lecithin and CMC. However, our research expands on this foundation by including other
emulsifiers, namely sucrose fatty acid esters and MDGs, and employing a distinct mouse model. Our
study offers novel insights, including the diverse impacts of different emulsifiers on health-related
markers and the examination of bacteria–mucus layer interactions. We have explicitly highlighted
the original contributions of our study in the revised manuscript.

**Classification of emulsifiers:** We understand your perspective on the classification of
emulsifiers into hydrophilic and lipophilic categories. While the results might not yield significant
conceptual advancements within the context of this emulsifier classification, their relevance lies in
our intention to explore potential variations in metabolic effects between these categories. A recent
report has identified seven dietary emulsifiers with high exposure levels in humans, and we focused
our study on the top four emulsifiers with high exposure levels: MDGs, lecithin, CMC, and sucrose
fatty acid esters. Our classification of these emulsifiers was guided by their solubility in water, with
MDGs lacking this property. Consequently, MDGs were administered through the diet, whereas
lecithin, sucrose fatty acid esters, and CMC were administered through the drinking water.

**Study's scope regarding international classification of dietary emulsifiers:** We value your
input concerning the scope of our rationale regarding international classification of emulsifiers. We
concur that this topic is better suited for journals dedicated to public health and food safety. Thus, we
have removed Figure 1, which presents data related to international classifications of dietary
emulsifiers, from the main figures and added it to the revised Supplementary Information as
Supplementary Figure 1. This modification ensures that the focus of our study is strongly aligned
with its biological findings and implications.

Specific comments:

Line1: The title ‘through the lens of gut microbiota’ is not mechanistic and is like a title for review
articles. This should be corrected.

**Response:**

Thank you for your feedback. The original title “Reassessing the Safety of Dietary Emulsifiers
Through the Lens of Gut Microbiota” has been modified in the revised manuscript as follows:

Lines 1 to 2: “Hydrophilic and lipophilic dietary emulsifiers promote metabolic disorders and
intestinal microbiota dysbiosis”

Line103-105: As above the effect of emulsifiers on metabolic syndrome has been already reported.
The rationale for reassessing the safety of emulsifiers is not clear. The authors should show the
mechanism how the differences between hydrophilic and hydrophobic emulsifiers affect the
metabolic outcome.

**Response:**

Thank you for your comments. At the beginning of the study, we categorized emulsifiers as
hydrophilic and lipophilic because MDGs could not be dissolved in water, unlike the remaining three
emulsifiers. It is essential to emphasize that the primary objective of our study was to investigate the
individual and collective effects of these emulsifiers on the gut microbiota composition and metabolic
biomarkers, rather than to conduct direct head-to-head comparisons between the effects of
hydrophilic and lipophilic emulsifiers, and determine their different underlying mechanisms. We
have clarified this in the revised manuscript.

Lines 102 to 105: “Moreover, at the beginning of the study, it became evident that MDGs
could not be dissolved in water, unlike the remaining emulsifiers. Consequently, we categorized the
selected emulsifiers as hydrophilic and lipophilic to better accommodate their distinct properties and
experimental requirements.”

Lines 426 to 433: “Moreover, we utilized two different forms of emulsifier administration,
namely drinking water and dietary supplementation. This approach may have contributed to the
observed differences in outcomes, making it challenging to directly compare the effects of
hydrophilic and lipophilic emulsifiers. In future studies aiming to compare the effects of these two
types of emulsifiers, both should be administered through dietary supplementation to minimize
potential sources of variability. Additionally, conducting a broader comparison including various
types of hydrophilic and lipophilic emulsifiers could yield more comprehensive and valuable
insights.”

Line486: The authors used ten times daily exposure in humans for LEC and SUC. Is this based on
the average exposure? Is there any previous evidence?

**Response:**

Thank you for your pertinent questions. In our study, we selected lecithin and sucrose fatty
acid ester doses based on the respective estimated dietary exposure levels in humans (lecithin, 61
653 mg/kg bw/day; sucrose fatty acid esters, 9 mg/kg bw/day)¹, ensuring the relevance of our findings to
654 human emulsifier consumption patterns. The dietary exposure levels of these emulsifiers in humans
were then translated to equivalent mouse doses⁴⁰, and these doses were increased 10 times. Thus,
mice were administered 7523.3 and 1110 mg/kg bw/day of lecithin and sucrose fatty acid esters,
respectively. We acknowledge that using such translation method involves certain assumptions and
limitations. However, this approach aligns with established practices in the field and serves as a
starting point for investigating the effects of these emulsifiers on mice in a controlled experimental
setting.

Line501: In the MDG diet, soybean oil was replaced with MDG. The authors should add the table
demonstrating nutrition facts in the MDG diet together with AIN-93M diet.

**Response:**

Thank you for your valuable suggestion. Accordingly, we have included a table presenting
the nutrition facts of both the MDG and AIN-93M diets in the revised Supplementary Information as
Supplementary Figure 7 (see below).

D10012M and New Diet

AIN-93M Mature Rodent Diet and Same with Soybean Oil Replaced by MDG/kg

Product #	D10012M		New Diet	
	gm%	kcal%	gm%	kcal%
Protein	14	15	14	15
Carbohydrate	73	76	73	76
Fat	4	9	4	9
Total		100		100
kcal/gm	3.8		3.8	
Ingredient	gm	kcal	gm	kcal
Casein	140	560	140	560
L-Cystine	1.8	7.2	1.8	7.2
Corn Starch	495.692	1982.768	495.692	1982.768
Maltodextrin 10	125	500	125	500
Sucrose	100	400	100	400
Cellulose, BW200	50	0	50	0
Soybean Oil	40	360	0	0
t-Butylhydroquinone	0.008	0	0.008	0
Mineral Mix S10022M	35	0	35	0
Vitamin Mix V10037	10	40	10	40
Choline Bitartrate	2.5	0	2.5	0
Mono and Diglycerides (MDG)	0	0	40	360
Total	1000	3850	1000.00	3850

**Supplementary Figure 7. Nutrition facts of the AIN-93M and new (MDG) diets (Research Diet**
**Inc.).**

Figure 1: Although the contents of this figure could be interesting, the data seems not to be original.
Instead, the authors summarized the classifications of emulsifiers in each organization, and they
nicely re-constructed the data for ref#13 and #9. It could be a main figure in public health-type review
articles. However, it should go to the supplementary table in the original article in biology journals.

**Response:**

Thank you for your suggestion. Accordingly, we have moved the original Figure 1 to the
revised Supplementary Information, and it is now labeled Supplementary Figure 1.

**References**

[revised manuscript text omitted]

Reviewers' comments:

Reviewer #1 (Remarks to the Author):

The authors have been highly responsive to the previous critiques and have made substantive and appropriate changes, as suggested. I believe that the manuscript is now ready for publication. The work will make an impact in the field with the important message that emulsifying food additives need to be reviewed individually and much more carefully.

Reviewer #2 (Remarks to the Author):

In this manuscript, the authors investigated the effects of hydrophilic dietary emulsifiers (LEC, SUC, CMC) and lipophilic dietary emulsifiers (MDG) on intestinal microbiota and their association with obesity and metabolic diseases, including disturbances in intestinal microbiota composition, alterations in mucous layer integrity, changes in intestinal permeability, and translocation of gut-derived LPS into the circulation system. Furthermore, they provide a comprehensive summary and discussion on the safety aspects of these emulsifiers. The manuscript is well-written and presents some innovative findings. However, there are still certain limitations that need to be addressed.

1. Only one lipophilic emulsifier, MDG, was studied in this manuscript, and its typicality remains unclear, thus limiting the generalizability of the conclusions drawn. It is recommended to explicitly highlight MDG in both the title and body rather than providing vague descriptions as a lipophilic emulsifier. Alternatively, the authors could enhance the comprehensiveness of the study by including more experiments involving lipophilic emulsifiers.

2. Line 256, the question is how to ensure that the intake of each mouse was limited to 85 kcal/week. Please elaborate.

3. After week 17 of the intervention, mice exhibited weight gain and dysregulated insulin homeostasis in this study. It remains unclear whether the disrupted insulin homeostasis is directly attributed to the emulsifier or solely a consequence of weight gain. To reinforce the validity of conclusions drawn, additional relevant experiments should be conducted and experimental design should be enhanced.

4. Did the positive control CMC employ the same dosage as LEC and SUC, that were 10 times the respective daily exposure levels in humans? If not, it would be preferable to minimize the comparison among different emulsifiers, such as the "Among the hydrophilic emulsifiers, sucrose fatty acid esters and CMC showed greater effects on health-related biomarkers than lecithin." mentioned in line 336. This is because even though they are hydrophilic emulsifiers, there is no assurance that their efficacy remains unaffected by factors like dosage. Instead, it would be more rigorous to discuss them individually.

5. The author states in line 498 of the METHODS section that "The mice were sacrificed after a 14-week experiment." However, there seems to be an inconsistency as the flow chart in Figure 3a depicts a 14-week intervention. Line 257, the author mentioned the intervention lasting 17 weeks, but described the body weight change at 14 weeks. This appears to be an error on the part of the authors. Additionally, it is worth noting that only data from 12 weeks are presented in line 260 and corresponding to Figure 3d,e. It would be helpful if you could explain why these time nodes are not

consistent.

Reviewer #3 (Remarks to the Author):

While the authors explained the rationale of their investigation in a rebuttal letter, the reviewer cannot see the new experimental data to address the lack of originality. In fact, the authors conclude that hydrophilic and lipophilic emulsifiers have similar effects on the read outs (e.g. dysbiosis, body weight) as the authors listed in the cartoon in Fig. 6.

For example, the authors again failed to address how the differences between hydrophilic and hydrophobic emulsifiers affect the metabolic outcome. Rather than that the authors explained that: It is essential to emphasize that the primary objective of our study was to investigate the individual and collective effects of these emulsifiers on the gut microbiota composition and metabolic biomarkers, rather than to conduct direct head-to-head comparisons between the effects of hydrophilic and lipophilic emulsifiers, and determine their different underlying mechanisms. Considering the fact that emulsifiers were conventionally classified, based on its nature as hydrophilic and lipophilic by using Hydrophile- Lipophile Balance Value (HLB value system), what is the rationale for categorizing of hydrophilic and lipophilic emulsifiers in this study?

Point-by-point response to reviewers' comments

Reviewer #1 (Remarks to the Author):

The authors have been highly responsive to the previous critiques and have made substantive and appropriate changes, as suggested. I believe that the manuscript is now ready for publication. The work will make an impact in the field with the important message that emulsifying food additives need to be reviewed individually and much more carefully.

Response:

We appreciate the positive feedback and are glad that the revisions have addressed previous critiques. Thank you for acknowledging the potential impact of our work in emphasizing the need for a careful and individual review of emulsifying food additives in the field.

**Reviewer #2 (Remarks to the Author):**

In this manuscript, the authors investigated the effects of hydrophilic dietary emulsifiers
(LEC, SUC, CMC) and lipophilic dietary emulsifiers (MDG) on intestinal microbiota and their
association with obesity and metabolic diseases, including disturbances in intestinal microbiota
composition, alterations in mucous layer integrity, changes in intestinal permeability, and
translocation of gut-derived LPS into the circulation system. Furthermore, they provide a
comprehensive summary and discussion on the safety aspects of these emulsifiers. The manuscript is
well-written and presents some innovative findings. However, there are still certain limitations that
need to be addressed.

**Response:**

We express gratitude for your valuable feedback on our manuscript. In response to your
comments and suggestions, we have carefully revised the manuscript to address the highlighted issues
and enhance its overall content.

1. Only one lipophilic emulsifier, MDG, was studied in this manuscript, and its typicality remains
unclear, thus limiting the generalizability of the conclusions drawn. It is recommended to explicitly
highlight MDG in both the title and body rather than providing vague descriptions as a lipophilic
emulsifier. Alternatively, the authors could enhance the comprehensiveness of the study by including
more experiments involving lipophilic emulsifiers.

**Response:**

Thank you for your valuable feedback on our manuscript. We have carefully considered your
suggestions and have made revisions accordingly. In response to your recommendation, we have
revised the manuscript title from “Hydrophilic and lipophilic dietary emulsifiers promote metabolic
disorders and induce intestinal microbiota dysbiosis” to “Common dietary emulsifiers promote
metabolic disorders and intestinal microbiota dysbiosis.” Additionally, we have explicitly highlighted
MDG in the subtitle of the results section and throughout the body of the manuscript, instead of using
vague descriptions such as “lipophilic emulsifier.” This modification aims to improve the clarity and
specificity of our findings, enhancing the overall comprehensibility of the study.

2. Line 256, the question is how to ensure that the intake of each mouse was limited to 85 kcal/week.

Please elaborate.

**Response:**

We have explained regarding each mouse was limited to approximately 85 kcal/week in the
Method section. “Mice were limited to 85 kcal/week¹, receiving 12.14 kcal/mouse (approximately
3.20 g/mouse) daily.” Given that the mice were housed in multiple cages per group (n = 3–4/cage),
our ability to precisely monitor the dietary intake of each individual mouse is constrained.
Consequently, we can approximate that each mouse received a diet of approximately 85 kcal/week.
Thus, we revised the sentence (lines: 521-522) from “mice were limited to 85 kcal/week¹” to “mice
were limited to approximately 85 kcal/week¹”.

3. After week 17 of the intervention, mice exhibited weight gain and dysregulated insulin homeostasis
in this study. It remains unclear whether the disrupted insulin homeostasis is directly attributed to the
emulsifier or solely a consequence of weight gain. To reinforce the validity of conclusions drawn,
additional relevant experiments should be conducted and experimental design should be enhanced.

**Response:**

We have conducted an additional experiment to investigate the direct effect of emulsifiers on
insulin resistance using an *in vitro* cellular insulin-resistance (IR) model in 3T3-L1 adipocytes. This
experiment involved measuring 2-deoxyglucose uptake. Furthermore, we have made revisions to the
main text in both the results and discussion sections as follows:

Results section (lines 142 to 152): “Given the observed insulin resistance (IR) in the
emulsifier-fed mice, we proceeded to investigate whether these emulsifiers directly influence glucose
homeostasis by using insulin resistance *in vitro* model with 3T3-L1 adipocytes. The doses of
emulsifiers administered to mice were extrapolated to doses for the IR cellular model (at ratios of
1:100 and 1:1000 for each emulsifier) (Supplementary Fig. 4). Lecithin and sucrose fatty acid ester
significantly reduced 2-deoxyglucose uptake at 30 min ($P=0.0010$ and $P<0.0001$, respectively),
suggesting insulin resistance induced by these emulsifiers. At 60 min, lecithin, sucrose fatty acid
ester, and carboxymethylcellulose (CMC) exhibited reductions in 2-deoxyglucose uptake ($P<0.0001$)
(Fig. 1n-o). However, the effect size observed with CMC was relatively small compared to the other
two emulsifiers. Hence, it may be inferred that lecithin and sucrose fatty acid ester directly contribute
to IR whereas IR induced by CMC may be related to the weight gain.”

Discussion section (lines 357 to 363): “According to findings from our *in vitro* study, lecithin
and sucrose fatty acid ester directly enhanced insulin resistance in the 3T3-L1 adipocytes, whereas
CMC had a relatively smaller effect. In contrast, our animal study revealed notable increases in fasting
glucose, serum insulin, and HOMA-IR in the CMC group compared to other emulsifiers used in this
study. Furthermore, a significant body weight gain was observed in the CMC group. These findings
suggest a potential association between the heightened insulin resistance in the CMC group and the
observed increase in body weight.”

**Figure 1. Dietary emulsifiers, including sucrose fatty acid esters and CMC, had an adverse**
 **effect on obesogenic and metabolic biomarkers, inducing insulin resistance *in vivo* and *in vitro*.**

**a)** Animal experimental design, **b)** changes in weight gain, **c)** changes in weight gain at 17th week,
 **d)** changes in relative fat mass, **e)** changes in relative lean mass, **f)** serum total cholesterol levels, **g)**
 total triglyceride levels, **h)** oral glucose tolerance test (OGTT) curve, **i)** area under the curve (AUC)
 of OGTT, **j)** serum fasting glucose levels, **k)** serum insulin levels, and **l)** homeostatic model
 assessment for insulin resistance (HOMA-IR), **m)** experimental design of *in vitro* cellular insulin-
 resistance (IR) model in 3T3-L1 adipocytes, **n)** 3T3-L1 adipocytes 2-deoxyglucose uptake at 30 min,
 and **o)** 60 min. Mice were supplemented with or without different emulsifiers in drinking water for

17 weeks. Dot plots are expressed as the mean \pm s.d. (n=14–15 per group). Insulin resistance (IR)
 was induced in 3T3-L1 adipocytes using dexamethasone for 72 hours. Emulsifiers, including LEC at
 concentrations of 0.2 mg/mL (1:1000) and 2 mg/mL (1:100), SUC at concentrations of 0.03 mg/mL
 (1:1000) and 0.3 mg/mL (1:100), and CMC at concentrations of 0.1 mg/mL (1:1000) and 1 mg/mL
 (1:100), were also introduced into the experimental setup. Statistical analyses were performed using
 one-way ANOVA with Tukey's range test for comparisons shown as exact P-values. *, P<0.05 and
 **, P<0.01. The 2-deoxyglucose uptake data were analyzed using one-way ANOVA with Dunnett's
 multiple comparisons test, comparing against the IR group. CON: control group; LEC: lecithin group;
 SUC: sucrose fatty acid esters group; CMC: carboxymethylcellulose group.

Emulsifier	The dose of emulsifiers administered to mice mimics human exposure levels (10X)		Emulsifier doses for 3T3-L1 cell (Translation from mouse dose)	
	mg/kg bw/day	mg [for 28 g mouse/day]	1:100 mg/mL	1:1000 mg/mL
LEC  Lecithin	7523	210.644	2.1064	0.2106
SUC  Sucrose ester	1110	31.08	0.3108	0.0311
CMC  Carboxymethylcellulose	3486 (Calculated based on actual intake)	97.6	0.9760	0.0976

**Supplementary Figure 4. The conversion of emulsifier doses administered to mice into doses**
 **suitable for the insulin resistance model in 3T3-L1 adipocytes.**

4. Did the positive control CMC employ the same dosage as LEC and SUC, that were 10 times the
respective daily exposure levels in humans? If not, it would be preferable to minimize the comparison
among different emulsifiers, such as the “Among the hydrophilic emulsifiers, sucrose fatty acid esters
and CMC showed greater effects on health-related biomarkers than lecithin.” mentioned in line 336.
This is because even though they are hydrophilic emulsifiers, there is no assurance that their efficacy
remains unaffected by factors like dosage. Instead, it would be more rigorous to discuss them
individually.

**Response:**

We appreciate the reviewer's insightful comment. The positive control CMC did not employ
the same dosage as LEC and SUC. To enhance the clarity and rigor of our study, we revised the
manuscript to discuss each emulsifier individually, avoiding direct comparisons. This adjustment will
provide a more accurate and nuanced analysis of the effects of different emulsifiers. Thank you for
your valuable suggestion

5.The author states in line 498 of the METHODS section that "The mice were sacrificed after a 14-
117 week experiment." However, there seems to be an inconsistency as the flow chart in Figure 3a depicts
a 14-week intervention. Line 257, the author mentioned the intervention lasting 17 weeks, but
described the body weight change at 14 weeks. This appears to be an error on the part of the authors.
Additionally, it is worth noting that only data from 12 weeks are presented in line 260 and
corresponding to Figure 3d,e. It would be helpful if you could explain why these time nodes are not
consistent.

**Response:**

We sincerely acknowledge the observation made by the reviewer and appreciate the
opportunity to address the discrepancy in our manuscript. Upon careful review, we acknowledge that
there was indeed an error in the reporting of the intervention duration. The correct duration of the
intervention is 14 weeks. We have promptly rectified the inconsistency in the manuscript, specifically
in Line 264, to accurately reflect the intervention duration as follows: "After intervention with MDG
for 14 weeks, no significant differences with respect to changes in body weight at 14 weeks were
observed (Fig. 3b, c)."

The data for relative lean mass and fat mass, as depicted in Figure 3d and e, were collected at
four time points (0, 4, 8, and 12 weeks). The lack of data at 14 weeks is attributed to the use of the
Minispec LF50 TD-NMR body composition analyzer (Bruker, Billerica, MA, USA), as described in
the Methods section. Due to the external location of the machine, mice were transported to the Taiwan
Mouse Clinic (Taipei, Taiwan) for each time point. Following the last measurement (12 weeks), the
mice were brought back to our institute for subsequent sacrifice at 14 weeks.

**Reviewer #3 (Remarks to the Author):**

While the authors explained the rationale of their investigation in a rebuttal letter, the reviewer
cannot see the new experimental data to address the lack of originality. In fact, the authors conclude
that hydrophilic and lipophilic emulsifiers have similar effects on the read outs (e.g. dysbiosis, body
weight) as the authors listed in the cartoon in Fig. 6.

For example, the authors again failed to address how the differences between hydrophilic and
hydrophobic emulsifiers affect the metabolic outcome. Rather than that the authors explained that: It
is essential to emphasize that the primary objective of our study was to investigate the individual and
collective effects of these emulsifiers on the gut microbiota composition and metabolic biomarkers,
rather than to conduct direct head-to-head comparisons between the effects of hydrophilic and
lipophilic emulsifiers, and determine their different underlying mechanisms.

Considering the fact that emulsifiers were conventionally classified, based on its nature as
hydrophilic and lipophilic by using Hydrophile- Lipophile Balance Value (HLB value system), what
is the rationale for categorizing of hydrophilic and lipophilic emulsifiers in this study?

**Response:**

We appreciate the valuable feedback provided on our manuscript. In response to your
comments and suggestions, we have revised the manuscript to address the highlighted issues and
improve its overall content.

Regarding our previous conclusion that hydrophilic and lipophilic emulsifiers have similar
effects on the readouts, we acknowledge the limitation of drawing such a conclusion given that only
one lipophilic emulsifier, MDG, was studied in this manuscript. To rectify this, we have revised the
manuscript title from “Hydrophilic and lipophilic dietary emulsifiers promote metabolic disorders
and induce intestinal microbiota dysbiosis” to “Common dietary emulsifiers promote metabolic
disorders and intestinal microbiota dysbiosis.” Additionally, we have explicitly highlighted MDG in
the subtitle of the results section and throughout the body of the manuscript, instead of using vague
descriptions such as “lipophilic emulsifier.” This modification aims to improve the clarity and
specificity of our findings, enhancing the overall comprehensibility of the study.

In this revised manuscript, we have conducted an additional experiment to investigate the
effect of emulsifiers on insulin resistance using an in vitro cellular insulin-resistance (IR) model in
3T3-L1 adipocytes. Our results demonstrated that lecithin and sucrose fatty acid ester significantly
reduced 2-deoxyglucose uptake, whereas CMC had a lesser impact, indicating a direct disruption of
insulin homeostasis by these emulsifiers (Fig. 1m-o; the figure is presented at the end of the text).

Furthermore, we have made revisions to the main text in both the results and discussion
sections. In response to your query regarding the rationale for categorizing emulsifiers as hydrophilic

and lipophilic in our study, we acknowledge the conventional classification of emulsifiers based on
their nature using the Hydrophile-Lipophile Balance Value (HLB value system). However, we have
found through literature review that MDG has an approximate HLB value of 4, soy lecithin has an
approximate HLB value of 7, and sucrose fatty acid esters have an HLB value of 15. Emulsifiers
possessing lower HLB values exhibit greater solubility in oil (lipophilic), whereas those with higher
values demonstrate enhanced solubility in water (hydrophilic). Typically, emulsifiers with HLB
values ranging from 3 to 6 are considered lipophilic and are most effective in water-in-oil (w/o)
emulsions. Conversely, hydrophilic emulsifiers, characterized by HLB values between 10 and 18, are
optimal for oil-in-water (o/w) emulsions^{2, 3, 4}.

While CMC is a water-soluble polysaccharide and does not have a reported HLB value, it is
commonly used in food and pharmaceutical industries⁵. Given this, we have determined that the HLB
value system may not be suitable for classifying emulsifiers in our study. We wish to emphasize that,
in this study, emulsifiers were categorized into two distinct forms of administration: drinking water
and dietary supplementation. This categorization was based on the inability of MDGs to dissolve in
water, unlike the remaining three emulsifiers (lecithin, sucrose fatty acid ester, and CMC).

Therefore, we have revised the manuscript title and explicitly highlighted MDG throughout
the body of the manuscript, instead of using vague descriptions such as “lipophilic emulsifier.” This
modification aims to improve the clarity and specificity of our findings, enhancing the overall
comprehensibility of the study. We have also revised Figure 5 as depicted below.

Original version

Revised version

**Figure 1. Dietary emulsifiers, including sucrose fatty acid esters and CMC, had an adverse**
 **effect on obesogenic and metabolic biomarkers, inducing insulin resistance *in vivo* and *in vitro*.**

**a)** Animal experimental design, **b)** changes in weight gain, **c)** changes in weight gain at 17th week,
 **d)** changes in relative fat mass, **e)** changes in relative lean mass, **f)** serum total cholesterol levels, **g)**
 total triglyceride levels, **h)** oral glucose tolerance test (OGTT) curve, **i)** area under the curve (AUC)
 of OGTT, **j)** serum fasting glucose levels, **k)** serum insulin levels, and **l)** homeostatic model
 assessment for insulin resistance (HOMA-IR), **m)** experimental design of *in vitro* cellular insulin-
 resistance (IR) model in 3T3-L1 adipocytes, **n)** 3T3-L1 adipocytes 2-deoxyglucose uptake at 30 min,
 and **o)** 60 min. Mice were supplemented with or without different emulsifiers in drinking water for

17 weeks. Dot plots are expressed as the mean \pm s.d. (n=14–15 per group). Insulin resistance (IR)
was induced in 3T3-L1 adipocytes using dexamethasone for 72 hours. Emulsifiers, including LEC at
concentrations of 0.2 mg/mL (1:1000) and 2 mg/mL (1:100), SUC at concentrations of 0.03 mg/mL
(1:1000) and 0.3 mg/mL (1:100), and CMC at concentrations of 0.1 mg/mL (1:1000) and 1 mg/mL
(1:100), were also introduced into the experimental setup. Statistical analyses were performed using
one-way ANOVA with Tukey's range test for comparisons shown as exact P-values. *, P<0.05 and
**, P<0.01. The 2-deoxyglucose uptake data were analyzed using one-way ANOVA with Dunnett's
multiple comparisons test, comparing against the IR group. CON: control group; LEC: lecithin group;
SUC: sucrose fatty acid esters group; CMC: carboxymethylcellulose group.

**References**

- 1. Sohal, R. S. & Weindruch, R. Oxidative stress, caloric restriction, and aging. *Science* **273**,
59-63 (1996).
- 2. Miller, R. Emulsifiers: Types and uses. (2016).
- 3. Golodnizky, D. & Davidovich-Pinhas, M. The effect of the HLB value of sucrose ester on
physiochemical properties of bigel systems. *Foods* **9**, 1857 (2020).
- 4. Mitsubishi Chemical Corporation, M. C. Sugar ester P-1570. URL:
<https://www.mfc.co.jp/english/p1570.htm> (2023).
- 5. Costa, E. M. et al. Carboxymethyl cellulose as a food emulsifier: Are its days numbered?
*Polymers* **15**, 2408 (2023).

REVIEWERS' COMMENTS:

Reviewer #2 (Remarks to the Author):

The author provides a thorough response to the previous comments and includes additional experimental evidence. I believe the manuscript is almost ready for publication. However, I have observed an inconsistency in the 10th citation made by the author (Nature. 2015;519(7541):92-96.). It states that under the same intervention dosage, the emulsifier CMC group reduced the distance of bacteria from intestinal epithelium cells and altered their pro-inflammatory potential. This finding appears to contradict the conclusions drawn in this study.

A point-by-point response to reviewers' comments

Reviewer #2 (Remarks to the Author):

The author provides a thorough response to the previous comments and includes additional experimental evidence. I believe the manuscript is almost ready for publication. However, I have observed an inconsistency in the 10th citation made by the author (Nature. 2015;519(7541):92-96.). It states that under the same intervention dosage, the emulsifier CMC group reduced the distance of bacteria from intestinal epithelium cells and altered their pro-inflammatory potential. This finding appears to contradict the conclusions drawn in this study.

Response:

We express gratitude for your valuable feedback on our manuscript. Upon revisiting the 10th citation (Nature. 2015;519(7541):92-96), we acknowledge that the findings reported therein suggest a reduction in the distance of bacteria from intestinal epithelial cells and alterations in their pro-inflammatory potential under the same intervention dosage of the emulsifier CMC. However, it's important to note that our study's conclusions are based on a different experimental design and context, which may lead to variations in observed outcomes. We believe that this apparent contradiction underscores the complexity and multifaceted nature of gut microbiota responses to dietary interventions, wherein various factors such as duration of exposure, host physiology, and experimental conditions can influence outcomes. As such, we have discussed this inconsistency in the discussion section of the revised manuscript to provide clarity and ensure transparency for the readers.

Lines 425 to 431: “CMC and P80 promote low-grade inflammation, metabolic syndrome, and colitis in mice by inducing microbiota encroachment, altering bacteria composition, and increasing intestinal permeability and LPS levels¹. Our study demonstrated that CMC induced metabolic disorders and altered bacteria composition, but did not reduce the distances of the closest bacterial cells to IECs. The inconsistency in our findings is possibly attributed to differences in the number of measurements of the closest bacteria to IECs, duration of emulsifier treatment, and gut microbiota composition.”

**References**

- 1. Chassaing, B. et al. Dietary emulsifiers impact the mouse gut microbiota promoting colitis
and metabolic syndrome. *Nature* **519**, 92-96 (2015).
